# A single H/ACA small nucleolar RNA mediates tumor suppression downstream of oncogenic RAS

Mary McMahon[1†], Adrian Contreras[1†], Mikael Holm[2‡], Tamayo Uechi[1], Craig M Forester[1,3], Xiaming Pang[1], Cody Jackson[4], Meredith E Calvert[4], Bin Chen[5,6], David A Quigley[7], John M Luk[8], R Kate Kelley[9], John D Gordan[9], Ryan M Gill[10], Scott C Blanchard[2‡*], Davide Ruggero[1,11*]

[1]Helen Diller Family Comprehensive Cancer Center, Department of Urology, University of California, San Francisco, San Francisco, United States; [2]Department of Physiology and Biophysics, Weill Cornell Medicine, New York, United States; [3]Division of Pediatric Allergy, Immunology & Bone Marrow Transplantation, University of California, San Francisco, San Francisco, United States; [4]Gladstone Histology and Light Microscopy Core, Gladstone Institutes, San Francisco, United States; [5]Department of Pediatrics and Human Development, Michigan State University, Grand Rapids, United States; [6]Department of Pharmacology and Toxicology, Michigan State University, Grand Rapids, United States; [7]Helen Diller Family Comprehensive Cancer Center and Department of Epidemiology and Biostatistics, University of California, San Francisco, San Francisco, United States; [8]Arbele Corporation, Seattle, United States; [9]Helen Diller Family Comprehensive Cancer Center, Department of Medicine, University of California, San Francisco, San Francisco, United States; [10]Department of Pathology, University of California, San Francisco, San Francisco, United States; [11]Department of Cellular and Molecular Pharmacology, University of California, San Francisco, San Francisco, United States

*For correspondence:
scott.blanchard@stjude.org (SCB);
davide.ruggero@ucsf.edu (DR)

†These authors contributed equally to this work

Present address: ‡Department of Structural Biology, St Jude Children's Research Hospital, Memphis, United States

**Abstract** Small nucleolar RNAs (snoRNAs) are a diverse group of non-coding RNAs that direct chemical modifications at specific residues on other RNA molecules, primarily on ribosomal RNA (rRNA). SnoRNAs are altered in several cancers; however, their role in cell homeostasis as well as in cellular transformation remains poorly explored. Here, we show that specific subsets of snoRNAs are differentially regulated during the earliest cellular response to oncogenic RAS$^{G12V}$ expression. We describe a novel function for one H/ACA snoRNA, *SNORA24*, which guides two pseudouridine modifications within the small ribosomal subunit, in RAS-induced senescence in vivo. We find that in mouse models, loss of *Snora24* cooperates with RAS$^{G12V}$ to promote the development of liver cancer that closely resembles human steatohepatitic hepatocellular carcinoma (HCC). From a clinical perspective, we further show that human HCCs with low *SNORA24* expression display increased lipid content and are associated with poor patient survival. We next asked whether ribosomes lacking *SNORA24*-guided pseudouridine modifications on 18S rRNA have alterations in their biophysical properties. Single-molecule Fluorescence Resonance Energy Transfer (FRET) analyses revealed that these ribosomes exhibit perturbations in aminoacyl-transfer RNA (aa-tRNA) selection and altered pre-translocation ribosome complex dynamics. Furthermore, we find that HCC cells lacking *SNORA24*-guided pseudouridine modifications have increased translational miscoding and stop codon readthrough frequencies. These findings highlight a role for specific snoRNAs in safeguarding against oncogenic insult and demonstrate a functional link between H/ACA snoRNAs regulated by RAS and the biophysical properties of ribosomes in cancer.

DOI: https://doi.org/10.7554/eLife.48847.001

## Introduction

Non-coding RNAs (ncRNAs) encompass a large group of functionally diverse non-protein coding transcripts that are emerging as important regulators of biological processes (*Cech and Steitz, 2014*; *Esteller, 2011*). Small nucleolar RNAs (snoRNAs) are abundant, often intron-encoded, short ncRNAs classified based on specific sequence and secondary structure features (*Kiss, 2002*; *Matera et al., 2007*). The most well-characterized functions of snoRNAs relate to their roles in ribosome biogenesis, wherein structurally distinct C/D and H/ACA snoRNAs directly base pair to complementary regions of ribosomal RNA (rRNA) (*Filipowicz and Pogacić, 2002*). In doing so, C/D and H/ACA snoRNAs modulate the chemical landscape of the ribosome by directing ribonucleoprotein complexes to modify up to two hundred site-specific ribose methylations (*2'-O-Me*) and pseudouridine ($\Psi$) modifications, respectively (*Sloan et al., 2017*; *Watkins and Bohnsack, 2012*). Unlike nucleotide modifications performed by stand-alone RNA modifying enzymes, the function of the vast majority of RNA-directed modifications such as those guided by snoRNAs, remain poorly studied.

Recent discoveries have shown that dysregulations in ribosome activity and protein synthesis are hallmarks of many cancer types (*Freed et al., 2010*; *Marcel et al., 2013*; *Pelletier et al., 2018*; *Robichaud and Sonenberg, 2017*; *Sulima et al., 2017*; *Truitt and Ruggero, 2016*). Emerging evidence suggests that the expression and activity of snoRNAs is also altered in a variety of human diseases, including cancer (*Belin et al., 2009*; *Bellodi et al., 2013*; *Ferreira et al., 2012*; *Gong et al., 2017*; *Mei et al., 2012*; *Ronchetti et al., 2013*; *Sahoo et al., 2008*; *Valleron et al., 2012*; *Williams and Farzaneh, 2012*). SnoRNA expression profiles have also been proposed as 'predictors' of specific cancer subtypes and clinical outcomes (*Ronchetti et al., 2013*; *Valleron et al., 2012*). Altered snoRNA expression in a variety of human cancers open several questions as to how snoRNAs may be regulated downstream of key oncogenic drivers in human tumors. However, it has yet to be examined whether snoRNA dysfunction plays a direct causative role in specific stages of cancer progression. While a loss of individual snoRNAs in single-celled organisms appears to be compatible with life (*Lowe and Eddy, 1999*; *Ni et al., 1997*), the precise biological impact of distinct snoRNA-directed modifications within defined regions of the ribosome in cancer development remains poorly understood.

Here, we find that specific subsets of H/ACA snoRNAs, that mediate pseudouridine modifications, are selectively regulated upon activation of oncogenic RAS. Upon oncogenic insult, primary cells normally activate a tumor suppressive response to counteract cellular transformation, known as oncogene-induced senescence (OIS) (*Collado et al., 2007*). We show that loss of only one distinct RAS-induced snoRNA, *SNORA24* (or H/ACA snoRNA 24), leads to the bypass of OIS in a liver model of RAS-induced senescence in vivo. SNORA24, which mediates two distinct pseudouridine modifications in the small, 40S subunit of the ribosome is also decreased in human hepatocellular carcinoma (HCC). We further show that loss of *Snora24* cooperates with RAS$^{G12V}$ to promote the development of liver cancer in vivo resembling a subtype of HCC characterized by lipid deposition, with similar features, as described in human steatohepatitic HCC (SH-HCC) (*Salomao et al., 2010*). Changes in the biophysical properties of ribosomes in cancer cells arising from loss of specific snoRNAs has not previously been tested. Employing single-molecule Fluorescence Resonance Energy Transfer (smFRET) imaging, we demonstrate that ribosomes isolated from human HCC cells specifically lacking *SNORA24*-guided pseudouridine modifications within the small ribosomal subunit, differ in the efficiency of aminoacyl-transfer RNA (aa-tRNA) selection, consistent with downstream reductions in translation accuracy, and in the dynamic properties of the pre-translocation ribosome complex. These findings reveal an important function for specific snoRNAs in RAS-mediated oncogenic activity and provide evidence that ribosomes lacking site-specific rRNA modifications exhibit physical alterations in the translation machinery.

**eLife digest** Ribosomes are cellular machines responsible for translating the genetic code into proteins. Research has shown that changes in ribosome activity can contribute to healthy cells becoming cancerous. Ribosomes consist of proteins and other molecules known as ribosomal RNAs (or rRNAs for short). Before they can become part of a ribosome, another type of molecule called snoRNAs must modify new rRNAs. Indeed, many of the modifications that allow rRNAs to accurately translate genetic information into proteins are introduced by snoRNAs. As such, it is possible that changes to snoRNAs could contribute to the creation of cancerous cells by affecting how ribosomes operate.

To explore this possibility, McMahon, Contreras et al. examined snoRNAs in healthy cells grown in the laboratory that have been given pro-cancer signals, in cancer from mice, and in samples from human cancer patients. The investigation revealed that the activation of growth signals – a hallmark of many cancers – affects the abundance of some snoRNAs and changes the pattern of rRNA modifications they make on ribosomes. Reducing the levels of one such snoRNA called SNORA24 led mice to develop fatty liver cancer when combined with cancer-promoting growth signals. Analyzing why reducing the levels of SNORA24 led to liver cancer, McMahon, Contreras et al. found that ribosomes lacking rRNA modifications introduced by SNORA24 made more mistakes when producing proteins coded for by certain genes.

These results contribute to the view of ribosomes as a key hub for the transformation of healthy cells into cancer cells. Increasing the error rate of ribosomes could be a key driver in further changes that drive cancer development. This study also highlights the role of snoRNAs in responding to growth signals, particularly in cancer. These findings identify snoRNAs as new potential diagnostic factors and treatment targets.

DOI: https://doi.org/10.7554/eLife.48847.002

## Results

### Subsets of H/ACA snoRNAs are differentially regulated upon oncogenic insult

To investigate the role of H/ACA snoRNAs during the earliest cellular response to oncogene activation, we interrogated the expression of ~90 H/ACA snoRNAs in primary human skin fibroblasts using a snoRNA quantitative PCR (qPCR) array, in the context of oncogene-induced senescence (OIS) by expression of RAS$^{G12V}$ (*Pylayeva-Gupta et al., 2011*). While the levels of the vast majority of H/ACA snoRNAs appeared unchanged following RAS$^{G12V}$ expression in primary fibroblasts, we observed a dynamic change in the expression of 28 H/ACA snoRNAs compared to control cells (*Figure 1A* and *Figure 1—source data 1*) (for example *SNORA23*, *SNORA24*, *SNORA26*, *SNORA48*, and *SNORA67*). The majority of these H/ACA snoRNAs were predominately upregulated (FDR < 0.1), with the exception of 3 snoRNAs that were downregulated (*SNORA36C*, *SNORA53*, and *SNORA70B*) downstream of oncogenic RAS. The increase in H/ACA snoRNA levels upon RAS$^{G12V}$ expression is not associated with elevated global protein production. On the contrary, we observe a pronounced decrease in overall protein synthesis as a consequence of RAS-induced senescence detected by monitoring O-propargyl-puromycin (OPP) incorporation into newly synthesized proteins (*Figure 1B* and *Figure 1—figure supplement 1A*, top panel). This change in global protein synthesis is consistent with the cell cycle arrest that occurs upon induction of senescence. We next investigated whether changes in the expression of specific H/ACA snoRNAs is selective to RAS activation or whether it is similarly induced upon other oncogenic signals. Interestingly, downregulation of the tumor suppressor PTEN (*Figure 1—figure supplement 1A*, bottom panel), a known oncogenic event that also promotes OIS in primary cells, had no obvious effect on the expression of selective RAS-induced H/ACA snoRNAs in primary fibroblasts (*Figure 1—figure supplement 1B* and highlighted in *Figure 1A*). These findings suggest that distinct oncogenic lesions alter a unique snoRNA expression pattern during OIS.

To extend the broader implications of our findings to human cancer etiology, we analyzed H/ACA snoRNA expression in ~300 human cancers using previously published microarray gene

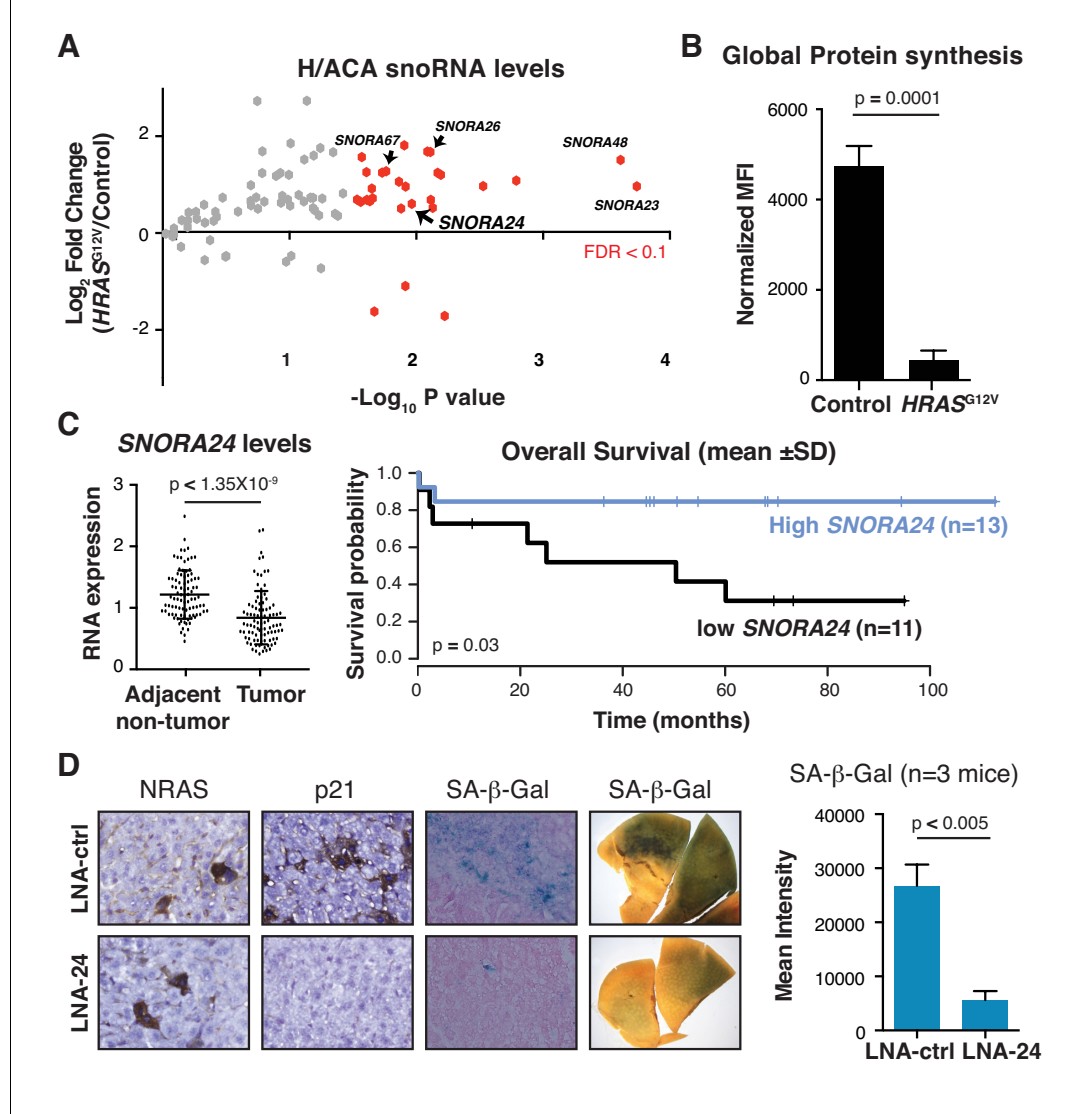

**Figure 1.** RAS-induced H/ACA snoRNAs are required for oncogene-induced senescence in vivo. (**A**) Volcano plot displays $Log_2$ fold change in H/ACA snoRNA levels 5 days following $HRAS^{G12V}$ expression in primary human skin fibroblasts measured by snoRNA qPCR array from three independent experiments. SnoRNAs in red exhibit statistically significant fold change in expression in $HRAS^{G12V}$ expressing cells compared to controls ($p<0.05$, unpaired Student's t-test or FDR < 0.1). SnoRNAs highlighted with labels were independently validated as shown in **Figure 1—figure supplement 1B**. (**B**) Graph illustrates mean ± SD mean fluorescent intensity (MFI) of the amount of de novo protein synthesis in primary human fibroblasts 5 days following expression of $HRAS^{G12V}$ compared to control treated cells by measuring OPP incorporation into newly synthesized protein from three independent experiments. Statistical analysis was performed using an unpaired Student's t-test, $p=0.0001$. (**C**) Analysis of $SNORA24$ levels in HCC specimens compared to adjacent non-tumor tissue of 91 HCC patients (GSE25097) (paired Student's t-test, $p<1.35\times10^{-9}$) (left panel) and Kaplan-Meier curve showing overall survival of HCC patients with high or low $SNORA24$ levels (mean ± 1 SD of $SNORA24$ levels) (right panel). Statistical significance was calculated using the log-rank test, with $p=0.03$. (**D**) Representative image for NRAS, p21, and SA-β-Gal staining in liver sections or resected liver lobes (SA-β-gal wholemount staining) 6 days following delivery of $NRAS^{G12V}$ and treated with control LNA (LNA-ctrl) or LNA targeting $Snora24$ (LNA-24). Graph shows mean ± SD mean intensity of SA-β-gal staining in liver from mice treated with LNA-ctrl (n = 3 mice) or LNA-24 (n = 3 mice) 6 days following $NRAS^{G12V}$ expression. Statistical analysis was performed using an unpaired Student's t-test, $p=0.005$.

DOI: https://doi.org/10.7554/eLife.48847.003

The following source data and figure supplements are available for figure 1:

**Source data 1.** H/ACA snoRNA levels upon oncogenic HRAS expression.
DOI: https://doi.org/10.7554/eLife.48847.010

**Figure supplement 1.** Select RAS-induced snoRNAs are not altered upon PTEN reduction.
DOI: https://doi.org/10.7554/eLife.48847.004

**Figure supplement 2.** Altered expression of distinct H/ACA snoRNAs in human cancers.

*Figure 1 continued on next page*

*Figure 1 continued*

DOI: https://doi.org/10.7554/eLife.48847.005

**Figure supplement 2—source data 1.** H/ACA snoRNA levels in human cancer.

DOI: https://doi.org/10.7554/eLife.48847.006

**Figure supplement 3.** Association between H/ACA snoRNA levels and patient survival in HCC.

DOI: https://doi.org/10.7554/eLife.48847.007

**Figure supplement 4.** *SNORA24* is reduced in primary human HCC specimens.

DOI: https://doi.org/10.7554/eLife.48847.008

**Figure supplement 5.** Loss of *Snora24* cooperates with RAS to promote the development of HCC resembling human SH-HCC.

DOI: https://doi.org/10.7554/eLife.48847.009

expression datasets (*Hao et al., 2011*; *Jima et al., 2010*; *Kabbout et al., 2013*; *Skrzypczak et al., 2010*; *Zhang et al., 2012*) designed to capture small RNA species, such as snoRNAs, that are often excluded during RNA sequencing library preparation. By assessing the average expression level of individual H/ACA snoRNAs in cancer specimens and matched control samples from five major tumor types (lymphoma [*Jima et al., 2010*], pancreatic [*Zhang et al., 2012*], colon [*Skrzypczak et al., 2010*], lung [*Kabbout et al., 2013*], and liver [*Hao et al., 2011*]), we identified H/ACA snoRNAs displaying changes in expression greater than ± 20% ($p<10^{-5}$) in tumor samples compared to control samples (*Figure 1—figure supplement 2* and *Figure 1—figure supplement 2—source data 1*). In the case of liver cancer, we observed an upregulation of 4 snoRNAs (*SNORA14B, SNORA17, SNORA72, and SNORA81*) and decreased expression of 2 H/ACA snoRNAs (*SNORA24 and SNORA67*) ($p<10^{-5}$) in 91 primary HCC specimens compared to matched adjacent non-tumor hepatic tissue (*Figure 1C*, left panel, *Figure 1—figure supplement 3*, and *Figure 1—figure supplement 2—source data 1*). Interestingly, two H/ACA snoRNAs, *SNORA24* and *SNORA67*, identified in our expression analysis upon RAS$^{G12V}$-induced senescence (*Figure 1A* and *Figure 1—figure supplement 1B*), were significantly decreased in HCC specimens compared to matched adjacent non-tumor tissue ($p<1.35\times10^{-9}$ and $p<4.6\times10^{-7}$, respectively). These findings suggest that *SNORA24* and *SNORA67* may act as tumor suppressors downstream of the early steps of oncogenic activation and may therefore be lost or downregulated during tumor progression. We next probed the clinical significance of altered H/ACA snoRNA expression in the same liver cancer microarray gene expression dataset, where survival data was available from 91 patients (*Hao et al., 2011*; *Kan et al., 2013*). After separating HCC patients into high or low snoRNA expression using mean ±1 standard deviation (SD) of snoRNA levels as a cutoff point, Kaplan-Meier curve analysis indicated that low levels of *SNORA24* were associated with worse overall survival compared to patients with high *SNORA24* expression (p=0.03, log-rank test) (*Figure 1C*, right panel). Interestingly, other snoRNAs assessed (including *SNORA17*, *SNORA67*, and *SNORA72*) did not demonstrate evidence for an association between expression levels and patient survival (*Figure 1—figure supplement 3A–C*, right panels). Importantly, we confirmed that *SNORA24* was decreased in an independent patient cohort consisting of primary matched tumor and adjacent non-tumor tissue from 13 HCC patients (*Supplementary file 1*, samples 1–13) using qPCR (*Figure 1—figure supplement 4A*). Similar to our observations from microarray gene expression analysis, *SNORA24* was dramatically decreased in all HCC tumor specimens compared to non-tumor adjacent tissue by qPCR (p<0.0001) (*Figure 1—figure supplement 4A*). In the same matched tumor and adjacent non-tumor tissue, we detected no change in the expression of the *SNORA24* host gene, *SNHG8* (Small Nucleolar RNA Host Gene 8) (*Figure 1—figure supplement 4B*). Together, these findings suggest that SNORA24 may exert a tumor suppressor role in HCC.

## Role for *Snora24* in the initiation and maintenance of RAS-driven HCC

As *SNORA24* was significantly decreased in HCC and low *SNORA24* levels were associated with poor patient survival, we next sought to assess the tumor suppressor activity of *Snora24* in liver cancer development in vivo. To this end, we employed a previously described mouse model of oncogenic RAS-induced senescence (*Carlson et al., 2005*; *Kang et al., 2011*) whereby hydrodynamic tail vein injection allows stable delivery of oncogenic RAS to hepatocytes using the sleeping beauty (SB) transposase system (*Figure 1—figure supplement 5A*). This model permits mosaic expression of RAS$^{G12V}$ in the mouse liver (SB(+)*NRAS*$^{G12V}$ mice) which activates an anti-tumor program (OIS) to

halt tumor development (*Kang et al., 2011*). We confirmed that RAS$^{G12V}$ promoted senescence in vivo after 6 days in SB(+)*NRAS*$^{G12V}$ mice livers by measuring two well-characterized markers of senescence, senescence-associated (SA) β-galactosidase (*Bandyopadhyay et al., 2005*) and a cell cycle inhibitor p21 (*Brown et al., 1997*), both of which are activated downstream of mitogenic signals to halt pre-malignant cells (*Collado et al., 2005*; *Kuilman et al., 2010*) (*Figure 1D*, top row). To test whether *Snora24* was implicated in OIS in SB(+)*NRAS*$^{G12V}$ mice, we used tail vein injection of locked nucleic acid (LNA) to target *Snora24* (LNA-24) for degradation. In so doing, we observed a specific reduction in *Snora24* levels in the mouse liver employing LNA-24 compared to a non-targeting scrambled sequence LNA (LNA-ctrl) (*Figure 1—figure supplement 5B*), with no observed impact on the levels of the corresponding snoRNA host gene, *Snhg8* (*Figure 1—figure supplement 5C*). Treatment with LNA-ctrl or LNA-24 had no obvious effects on hepatic tissue architecture or function (data not shown). Strikingly, we observed a bypass of OIS upon *Snora24* reduction in this model, as evident by reduced p21 expression and a lack of β-galactosidase staining in liver tissue from LNA-24 treated SB(+)*NRAS*$^{G12V}$ mice compared to LNA-ctrl treated SB(+)*NRAS*$^{G12V}$ mice (*Figure 1D*, right panel, p<0.005, n = 3 mice per condition). Importantly, LNA-24 decreased pseudouridine modifications on 18S rRNA at positions U609 and U863 (*Figure 1—figure supplement 5D*), without impacting modifications at other sites on 18S rRNA (U105) or 28S rRNA (U1731) that not guided by *SNORA24* (*Figure 1—figure supplement 5E*).

Having identified that cancer-associated changes in *Snora24* lead to bypass of OIS in the context of oncogenic RAS expression in vivo, we next sought to determine whether reduction of this single H/ACA snoRNA was sufficient to promote tumor development in this model. To do this, we analyzed the liver phenotype of SB(+)*NRAS*$^{G12V}$ mice treated with either LNA-ctrl or LNA-24 for four months. In contrast to LNA-ctrl treated SB(+)*NRAS*$^{G12V}$ mice, which have normal livers and do not show any signs of HCC (*Figure 2A*, bottom left panel, representative image from n = 8 mice), we found that a reduction in *Snora24* cooperates with RAS$^{G12V}$ to promote the development of liver cancer in all LNA-24 treated SB(+)*NRAS*$^{G12V}$ mice examined (*Figure 2A*, bottom right panel, representative image from n = 8 mice). Pathological analysis of liver specimens revealed the presence of HCC in LNA-24 treated SB(+)*NRAS*$^{G12V}$ mice, with evidence of dramatic lipid accumulation and swollen hepatocytes suggestive of hepatocyte balloons, the pathology of which resembles human steatohepatitic HCC (SH-HCC) (*Figure 2B*). Interestingly, SH-HCC is a subtype of liver cancer characterized by increased fat accumulation, the etiology and genetic makeup of which is poorly understood (*Salomao et al., 2010*). The presence of lipids in liver tumors derived from LNA-24 treated SB(+)*NRAS*$^{G12V}$ mice was confirmed by Oil Red O (ORO) staining (*Figure 2C* and *Figure 2—figure supplement 1*, n = 3 mice, p=0.0107). Taken together, these findings indicate that loss of *Snora24* cooperates with oncogenic RAS to promote the development of HCC in vivo. These data also indicate that cancer-associated changes in *SNORA24*, identified in HCC patient expression data (*Figure 1C*), may play a direct role in HCC pathogenesis.

To investigate the importance of *Snora24* in established HCC, we turned to a genetically engineered mouse model of liver cancer driven by the expression of oncogenic *Kras*$^{G12D}$ in the mouse liver using albumin-cre (*Alb*-cre;*Kras*$^{G12D}$). In this genetically engineered mouse model, oncogenic RAS is ubiquitously expressed in the fetal liver and therefore these animals develop HCC within 8 months of age (*O'Dell et al., 2012*; *Xu et al., 2019*). In line with human HCC patient expression data (*Figure 1C* and *Figure 1—figure supplement 4A*), we found that in established liver tumors derived from *Alb*-cre;*Kras*$^{G12D}$ mice, *Snora24* was significantly decreased in tumor tissue compared to liver tissue from wild-type mice (*Figure 2D*, n = 3 mice per condition). To test the tumor suppressive activity of *Snora24* in a more aggressive liver cancer, we orthotopically injected primary tumor cells derived from *Alb*-cre;*Kras*$^{G12D}$ liver tumor (*Xu et al., 2019*), following CRISPR-Cas9 gene editing of the *Snora24* loci using two distinct single guide RNA (sgRNA) targeting *Snora24* (sgRNA-24) (*Figure 2—figure supplement 2A*), into the livers of wild-type mice. sgRNA-24 *Kras*$^{G12D}$ HCC cells exhibit reduced Snora24 expression (*Figure 2E*, right panel) with no obvious impact on the levels of the corresponding host gene (*Figure 2—figure supplement 2B*). Following orthotopic injection of control (Ctrl) *Kras*$^{G12D}$ HCC cells or sgRNA-24 *Kras*$^{G12D}$ HCC cells into the subcapsular region of the median liver lobe of wild-type mice, we monitored survival over a period of 4 weeks and found that *Snora24* reduction, decreased the overall survival of mice compared to controls (p=0.017) (*Figure 2E*, left panel n = 4 mice per arm). *Snora24* reduction in this mouse model of liver cancer significantly decreased *Snora24*-guided pseudouridine modifications on 18S rRNA (*Figure 2—figure*

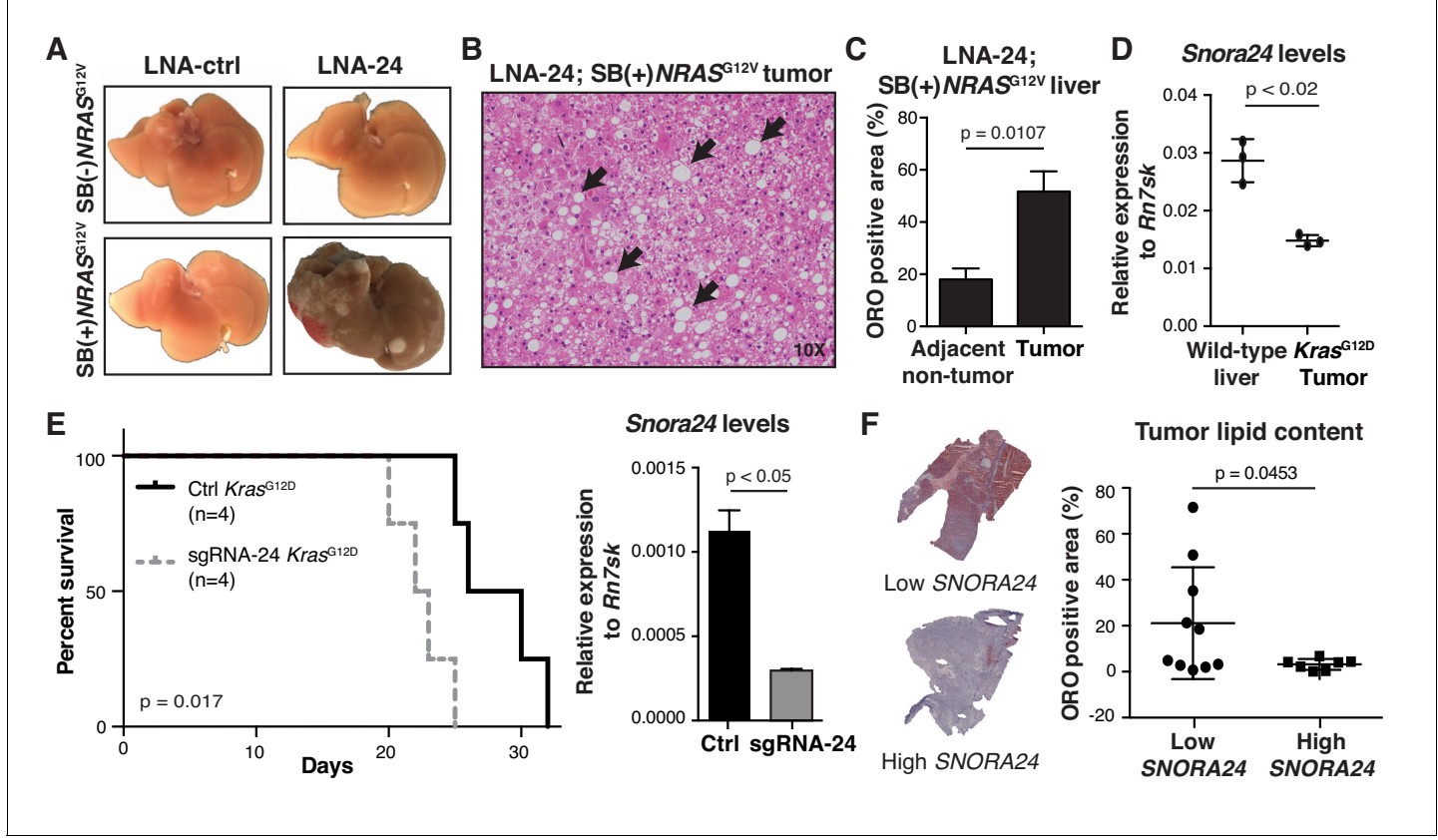

**Figure 2.** *Snora24* plays a role in the initiation and maintenance of RAS-driven hepatocellular carcinoma. (**A**) Representative images of explanted livers from control (SB(-)*NRAS*^G12V) or SB(+)*NRAS*^G12V mice treated with either LNA-ctrl or LNA-24. (**B**) H and E staining of a liver section from SB(+)*NRAS*^G12V mouse treated with LNA-24. Black arrows highlight the presence of fat droplets. (**C**) Graph shows mean ± SD percentage Oil Red O (ORO) positive area in liver tumor nodules and adjacent non-tumor tissue from n = 3 SB(+)*NRAS*^G12V mice treated with LNA-24. For each mouse liver section, the amount of ORO positive stain per total area from at least four distinct tumor and non-tumor regions (as determined by H and E staining) was measured (see Materials and methods and *Figure 2—figure supplement 1*). Statistical analysis was performed using a paired Student's t-test, p=0.0107). (**D**) Quantitative PCR (qPCR) analysis of *Snora24* levels in wild-type liver or age- and sex-matched liver tumors from *Alb*-cre;*Kras*^G12D mice. Graph shows mean ± SD *Snora24* expression normalized to the levels of *Rn7sk* from n = 3 mice per condition. Statistical analysis was performed using an unpaired Student's t-test, p<0.02. (**E**) Kaplan-Meier curves showing survival in male C57BL/6 wild-type mice following intrahepatic orthotopic injection of Ctrl *Kras*^G12D HCC cells (black line, n = 4 mice) and sgRNA-24 *Kras*^G12D HCC cells (gray dashed line, n = 4 mice), p=0.017, log-rank test (left panel). qPCR analysis of *Snora24* in Ctrl *Kras*^G12D and sgRNA-24 *Kras*^G12D HCC cells (right panel). Graph shows mean relative expression ± SD normalized to the levels of *Rn7sk* from three independent experiments. Statistical analysis was performed using an unpaired Student's t-test, *p < 0.05. (**F**) Representative image of ORO staining in HCC from a patient with high *SNORA24* (bottom) or low *SNORA24* (top) expression. Quantification of ORO stain in tissue sections from HCC patients dichotomized into high or low by identifying samples with *SNORA24* expression greater than ±one SD from the mean (n = 17 HCC specimens). Graph shows mean ± SD percentage Oil Red O (ORO) positive area present in HCC tissue specimens from patients with high or low *SNORA24* expression and statistical analysis was performed using an unpaired Student's t-test, p=0.0453.

DOI: https://doi.org/10.7554/eLife.48847.011

The following figure supplements are available for figure 2:

**Figure supplement 1.** Increased lipid content in tumor regions of LNA-24; SB(+)*NRAS*^G12V mice.
DOI: https://doi.org/10.7554/eLife.48847.012

**Figure supplement 2.** Reduction of *Snora24*-guided modifications in mouse *Kras*^G12D liver cancer cells using CRISPR-Cas9 gene editing.
DOI: https://doi.org/10.7554/eLife.48847.013

**Figure supplement 3.** Reduction of *SNORA24* in HuH-7 cells using CRISPR-Cas9 gene editing.
DOI: https://doi.org/10.7554/eLife.48847.014

*supplement 2C*) as shown using SCARLET (Site-specific Cleavage And Radioactive-labeling followed by Ligation-assisted Extraction and Thin-layer chromatography) (*Li et al., 2015*; *Liu and Pan, 2015*). These findings reveal a previously uncharacterized role for *Snora24* in the maintenance of RAS-driven HCC in vivo.

## Low SNORA24 levels in HCC is associated with higher lipid content

Given that loss of *Snora24* cooperates with oncogenic RAS to promote the development of liver cancer resembling human SH-HCC (*Figure 2A–C*), we sought to explore the clinical relationship between *SNORA24* expression and lipid content in human HCC. To do this, we took advantage of available primary tissue specimens from 62 HCC patients (*Supplementary file 1*). *SNORA24* expression was determined by qPCR and HCC specimens were stratified into high or low *SNORA24* using *SNORA24* levels greater than ±1 SD from the mean as a cut-off. Strikingly, we found that tumors with low *SNORA24* levels had higher lipid content compared to tumors with high SNORA24 levels using ORO staining (*Figure 2F*, p=0.0453, n = 17). These findings confirm a significant correlation between *SNORA24* levels and lipid content in HCC. We also found that *SNORA24* reduction by CRISPR-Cas9 gene editing in an established, well-differentiated human HCC cell line, HuH-7, (HuH-7 sgRNA-24) (*Figure 2—figure supplement 3A*), also enhanced lipid droplet formation compared to isogenic control (ctrl) HuH-7 cells (HuH-7 sgRNA-ctrl) (*Figure 2—figure supplement 3B*). Altogether, these findings highlight a previously unidentified connection between *SNORA24* and a particular feature of lipid accumulation associated with human SH-HCC.

## *SNORA24*-guided rRNA modifications influence biophysical properties of ribosomes

The impact of *SNORA24* and *SNORA24*-directed rRNA modifications on ribosome activity and global protein production remains unknown. *SNORA24* guides two pseudouridine modifications on the 18S rRNA component of the small (40S) ribosomal subunit (*Boccaletto et al., 2018*), one at position uridine (U) 609 (U609) within helix 18 (h18) and one at U863 at the base of expansion segment 6 (ES6) (*Figure 3A*, highlighted in red). While the ES6 modification is distal to known functional centers of the ribosome, the h18 modification is located within the functionally important, so-called '530 loop' region of the shoulder domain (*Figure 3A*, highlighted in pale green). The shoulder domain closes towards the body of the small ribosomal subunit during the process of aminoacyl-tRNA (aa-tRNA) selection, bringing residue G530 (G626 in human rRNA) into direct contact with the messenger RNA (mRNA) codon-tRNA anticodon pair such that it directly contributes to mRNA decoding (*Demeshkina et al., 2012*; *Fislage et al., 2018*; *Loveland et al., 2016*; *Ogle et al., 2001*; *Ogle et al., 2002*). We therefore next sought to investigate the role of these *SNORA24*-guided modifications in modulating ribosome function in HCC. To address this question, we employed a human HCC cell line, HuH-7, with a stable reduction in *SNORA24* (HuH-7 sgRNA-24) (*Figure 2—figure supplement 3A*) and quantified *SNORA24*-mediated pseudouridylation in h18 and ES6 using SCARLET (*Li et al., 2015*; *Liu and Pan, 2015*). As expected, SCARLET revealed the amount of uridine (U) and pseudouridine (Ψ) present at both residue 609 and 863 on 18S rRNA and confirmed diminished pseudouridine levels at both sites in HuH-7 sgRNA-24 cells compared to isogenic HuH-7 sgRNA-ctrl cells (>90% reduction) (*Figure 3B* and *Figure 3—figure supplement 1*). Interestingly, we did not detect any significant change in global protein production in cells lacking *SNORA24*-guided modifications as measured by OPP incorporation into newly synthesized protein (*Figure 3C*). Furthermore, we saw no detectable difference in the abundance of ribosome subunits or the abundance and distribution of polysome in HuH-7 sgRNA-24 cells compared to isogenic HuH-7 sgRNA-ctrl cells by sucrose gradient fractionation trace (*Figure 3D*). These results indicate that *SNORA24* and *SNORA24*-guided rRNA modifications are likely dispensable for ribosome biogenesis and global protein production in HCC and instead may harbor specific functions during translation.

To gain mechanistic insights into how Ψ609 and Ψ863 influence ribosome activity, we harnessed the power of smFRET imaging to capture the functional dynamics of ribosomes on mRNA (*Ferguson et al., 2015*; *Juette et al., 2016*). Ribosome transit along mRNA is a highly coordinated, multistep process involving the selection of the correct aa-tRNA in the ribosomal aminoacyl (A) site for each codon, peptide bond formation, and the translocation of the ribosome, codon-by-codon along the mRNA. We first imaged aa-tRNA selection or the process by which aa-tRNAs are decoded at the ribosomal A site to extend the nascent polypeptide by one amino acid using an established in vitro smFRET assay (*Blanchard et al., 2004*; *Ferguson et al., 2015*; *Geggier et al., 2010*). Briefly, translation initiation complexes (80S ICs) carrying Cy3-labeled Met-tRNA$^{fMet}$ in the peptidyl (P) site and displaying a UUC codon (encoding Phenylalanine (Phe)) in the ribosomal A site were formed from isolated ribosomal subunits purified from HuH-7 sgRNA-ctrl or sgRNA-24 cells (*Figure 4A*). The

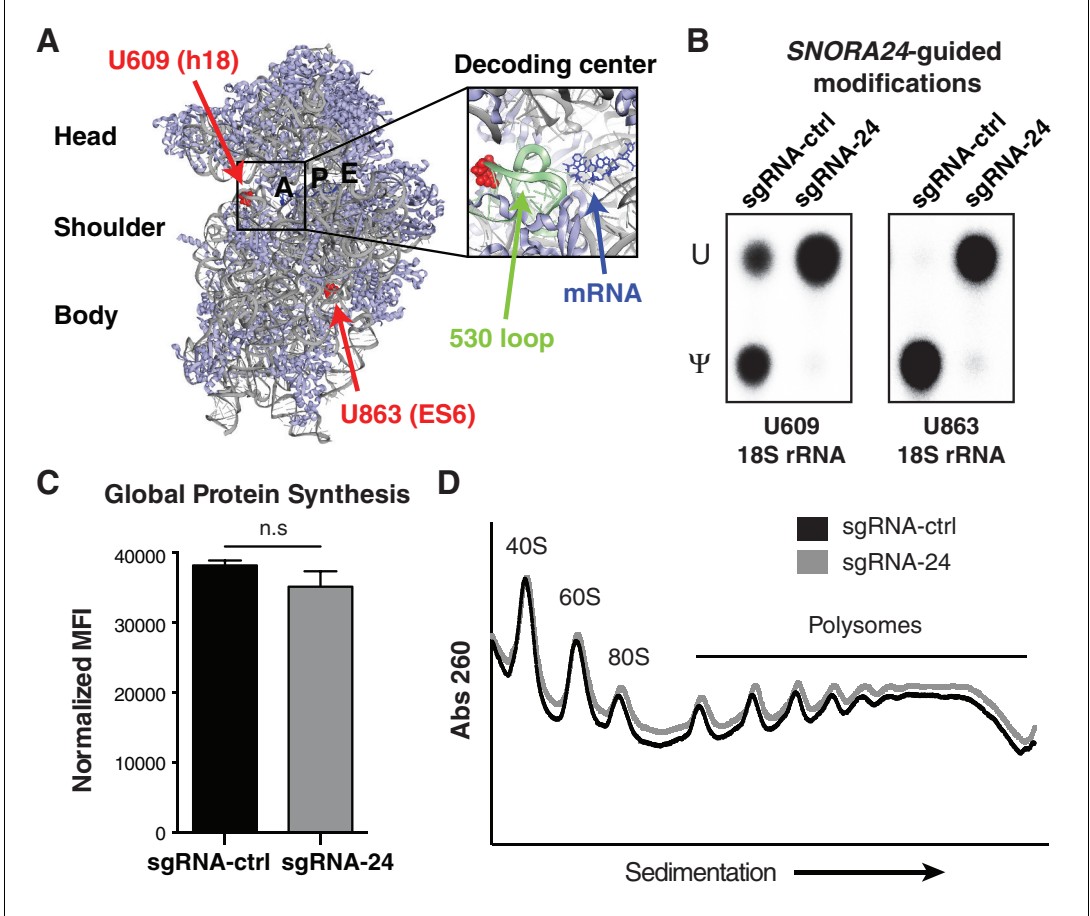

**Figure 3.** Loss of *SNORA24*-guided pseudouridine modifications does not impact global protein production in HCC cells. (**A**) *SNORA24* target residues U609 and U863 are highlighted (in red) on the structure of the mammalian 40S ribosomal subunit (Protein Data Bank (PDB) ID: 5LZS [*Shao et al., 2016*] and visualized with PyMOL). 18S rRNA is in gray, the mRNA in the decoding center is highlighted in blue, and the '530 loop' is highlighted in pale green. (**B**) Representative thin layer chromatography (TLC) of site-specific amounts of pseudouridine (Ψ) or uridine (U) present at position U609 and U863 on 18S rRNA using SCARLET in HuH-7 sgRNA-ctrl or sgRNA-24 cells. (**C**) Graph illustrates mean ± SD mean fluorescent intensity (MFI) of the amount of de novo protein synthesis in HuH-7 sgRNA-24 cells compared to HuH-7 sgRNA-ctrl cells by measuring OPP incorporation into newly synthesized protein from three independent experiments. Statistical analysis was performed using an unpaired Student's t-test, n.s = non significant. (**D**) Representative polysome profiles of HuH-7 sgRNA-ctrl (black line) and sgRNA-24 (gray line) cells. Lower molecular weight (MW) complexes (40S and 60S) are on the left side of the x axis and higher MW complexes (polysomes) are on the right side.

DOI: https://doi.org/10.7554/eLife.48847.015

The following figure supplement is available for figure 3:

**Figure supplement 1.** Loss of *SNORA24*-guided modifications in HuH-7 cells with reduced *SNORA24*.

DOI: https://doi.org/10.7554/eLife.48847.016

80S ICs were immobilized in a passivated microfluidic flow cell and imaged as described previously (*Juette et al., 2016*). Ternary complex, consisting of LD655-labeled Phe-tRNA$^{Phe}$, eukaryotic translation elongation factor 1A (eEF1A), and GTP, was then injected into the flow cell and the process of aa-tRNA selection was followed via the time-evolution of the FRET signal between the P-site tRNA and the aa-tRNA decoded at the A site, at a time resolution of 15 milliseconds (ms).

Ribosomes purified from both HuH-7 sgRNA-ctrl and sgRNA-24 cells, proceeded through the aa-tRNA selection mechanism, as expected (*Ferguson et al., 2015*; *Geggier et al., 2010*), via a step-wise progression through states of increasing FRET efficiency (*Figure 4B*). These FRET states correspond to initial Phe-tRNA$^{Phe}$ binding or codon recognition events (FRET ~0.2), from which aa-tRNAs can either rapidly dissociate or proceed to the GTPase-activated state (FRET ~0.4). GTP hydrolysis within the GTPase-activated state enables eEF1A to dissociate and allows aa-tRNAs to transition

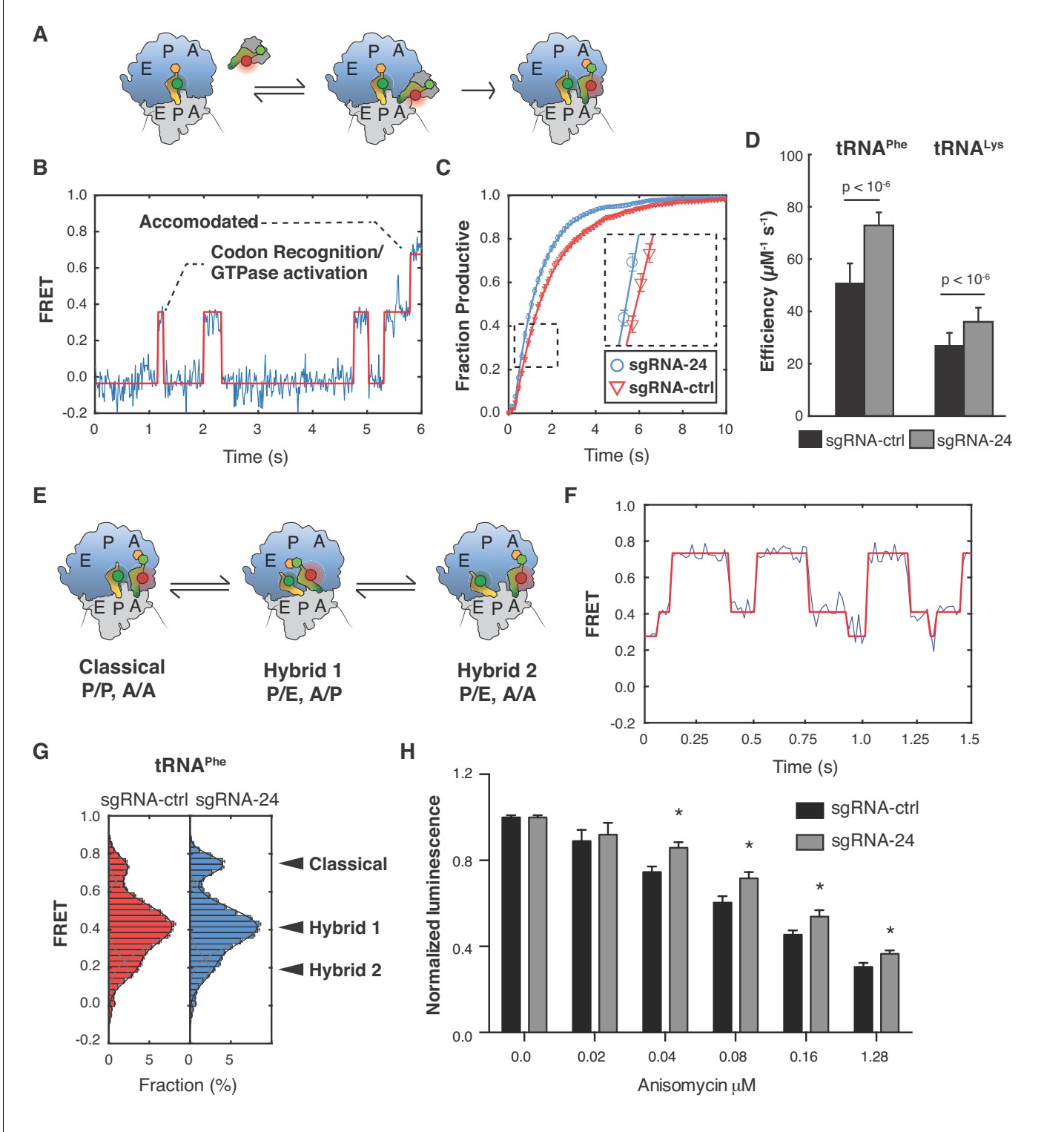

**Figure 4.** Ribosomes lacking *SNORA24*-guided modifications display alterations in aa-tRNA selection and pre-translocation complex dynamics. (A) Schematic representation of the reaction assayed in the smFRET experiments. Ternary complex consisting of eEF1A, GTP, and fluorescently labeled aa-tRNA (either LD655-tRNA^Phe or LD650-tRNA^Lys) binds to ribosomes, carrying Cy3 labeled Met-tRNA^fMet in the P site and displaying a cognate codon (UUC or AAA) in the A site, bringing the two dyes close enough for FRET. After binding, the ternary complex either dissociates or codon recognition, GTP hydrolysis, and subsequent dissociation of eEF1A takes place followed by accommodation of the CCA end of the tRNA into the peptidyl transferase center and subsequent peptide bond formation. These conformational changes within the decoding ribosome complex lead to a stepwise

*Figure 4 continued on next page*

*Figure 4 continued*

increase in FRET until stable accommodation of the tRNA occurs. (B) Representative smFRET trace of ribosome purified from HuH-7 sgRNA-ctrl cells displaying a UUC codon reacting with LD655-tRNA$^{Phe}$ containing ternary complex. Non-productive events are characterized by rapid fluctuations in FRET values between $0.2 \pm 0.075$ and $0.46 \pm 0.075$, followed by dissociation of the ternary complex and loss of FRET. Productive events are characterized by a stepwise progression from $0.2 \pm 0.075$, through several intermediate FRET values to a final FRET of $0.72 \pm 0.075$. The red line represents a hidden Markov-model idealization of the smFRET trace. (C) Cumulative distributions for ribosomes purified from HuH-7 sgRNA-ctrl (red line) and sgRNA-24 (blue line) cells displaying a UUC codon reacting with LD655-tRNA$^{Phe}$ containing ternary complex. Distributions were constructed from all recorded individual smFRET traces by estimating the number of productive events that occurred at each movie frame. The solid lines represent exponential functions fitted to the data. The error bars represent SEM for each data point. For simplicity, data from every tenth movie frame is shown. (D) Graph shows catalytic efficiency ($k_{cat}/K_M$) for LD655-tRNA$^{Phe}$ or LD650-tRNA$^{Lys}$ containing ternary complexes reacting on ribosomes purified from either HuH-7 sgRNA-ctrl or sgRNA-24 cells and displaying the respective cognate codons, UUC or AAA, in the A site. The error bars represent SEM for the estimated $k_{cat}/K_M$ values. Statistical analysis was performed using Welch's t-test, $p<10^{-6}$. (E) Schematic representation of the dynamics of ribosome pre-translocation complexes. Each ribosome can occupy three distinct conformational states, the classical state; with both tRNAs in classical binding conformations with their anticodon stems and CCA ends occupying corresponding binding sites on the small and large ribosomal subunits. The first hybrid state; with both tRNAs in hybrid binding conformations with their anticodon stems and CCA ends occupying different binding sites on the small and large subunits, and the second hybrid state where the A-site tRNA is in a classical binding conformation while the P-site tRNA is in a hybrid binding conformation. (F) Representative smFRET trace of a pre-translocation complex containing P-site bound tRNA$^{fMet}$ and A-site bound Met-Phe-tRNA$^{Phe}$. The highest FRET state corresponds to the classical state, the middle FRET state corresponds to the first hybrid state, and the lowest FRET state corresponds to the second hybrid state. The solid red line represents a hidden Markov-model idealization of the smFRET trace. (G) Histograms of FRET values attained by pre-translocation complexes from HuH-7 sgRNA-ctrl (red bars) and sgRNA-24 (blue bars) cells containing P-site bound tRNA$^{fMet}$ and A-site bound Met-Phe-tRNA$^{Phe}$. The solid lines represent fits of three gaussian functions (gray lines) to the data and their sum (black line). The error bars represent SEM for the mean count in each histogram bin. Statistical analysis was performed using Welch's t-test, $p<10^{-6}$. (H) Graph shows mean $\pm$ SD normalized luminescence from CellTiter-Glo Luminescent Cell Viability Assay 48 hrs post-treatment of HuH-7 sgRNA-ctrl (black bars) or sgRNA-24 (gray bars) cells with increasing concentrations of Anisomycin from three independent experiments. Statistical analysis was performed using an unpaired Student's t-test, *p<0.05.

DOI: https://doi.org/10.7554/eLife.48847.017

The following figure supplements are available for figure 4:

**Figure supplement 1.** Loss of *SNORA24*-guided modifications does not impact the dynamics of pre-translocation complexes containing A-site bound Met-Lys-tRNA$^{Lys}$.

DOI: https://doi.org/10.7554/eLife.48847.018

**Figure supplement 2.** Effects of specific ribosome targeting drugs on the growth of HCC cells lacking *SNORA24*-guided rRNA modifications.

DOI: https://doi.org/10.7554/eLife.48847.019

into a fully accommodated state, where Phe-tRNA$^{Phe}$ in the A site undergoes peptide bond formation (FRET ~0.7) (*Figure 4B*). We estimated the rate of peptide bond formation in this system and calculated the catalytic efficiency ($k_{cat}/K_M$) of tRNA selection (*Figure 4C and D*). In doing so, we found that ribosomes purified from HuH-7 sgRNA-24 cells were ~50% more efficient in aa-tRNA selection of tRNA$^{Phe}$ compared to ribosomes isolated from sgRNA-ctrl cells ($p<10^{-6}$, Welch's t-test) ($k_{cat}/K_M$ of $50 \pm 7$ $\mu M^{-1}$ $s^{-1}$ vs. $71 \pm 4$ $\mu M^{-1}$ $s^{-1}$, respectively) (*Figure 4D*). We further assessed ribosomes displaying a different codon, AAA (encoding Lysine (Lys)), in the A site reacting with ternary complex containing a cognate LD650 labeled Lys-tRNA$^{Lys}$ using the same assay. In this context, we found that ribosomes lacking $\Psi$609 and $\Psi$863 were also more efficient in selection of tRNA$^{Lys}$, however, the magnitude of the change was smaller than that observed for tRNA$^{Phe}$. Specifically, ribosomes from HuH-7 sgRNA-24 cells were ~30% more efficient in selection of tRNA$^{Lys}$ compared to their wild-type counterparts ($p<10^{-6}$, Welch's t-test) ($k_{cat}/K_M$ of $27 \pm 5$ $\mu M^{-1}$ $s^{-1}$ vs. $36 \pm 5$ $\mu M^{-1}$ $s^{-1}$) (*Figure 4D*) as opposed to the ~50% differences observed in the case of tRNA$^{Phe}$. These findings indicate that *SNORA24*-directed rRNA pseudouridylation acts to regulate decoding on actively translating ribosomes.

Ribosomes undergo large-scale conformational rearrangements during the elongation phase of translation (*Voorhees and Ramakrishnan, 2013*). For example, in pre-translocation ribosome complexes, deacylated tRNA within the P site and peptidyl-tRNA within the A site rapidly and dynamically exchange between so-called 'classical' and 'hybrid' states of tRNA binding (*Ferguson et al., 2015*). Classical (C: A/A, P/P) and hybrid states (H1: A/P, P/E; H2: A/A, P/E) can be distinguished by smFRET imaging based on their distinct FRET efficiency values (*Ferguson et al., 2015*) (*Figure 4E and F*). To assess the potential impacts of *SNORA24*-guided modifications on ribosome dynamics, we imaged pre-translocation complexes generated in the tRNA selection experiments described

above bearing Met-Phe-tRNA$^{Phe}$ in the A site. These analyses revealed that ribosomes lacking *SNORA24*-guided modifications exhibit a modest preference for classical tRNA configurations compared to control ribosomes (15 ± 1% vs. 20 ± 1% Classical, p<10$^{-6}$, Welch's t-test) (*Figure 4G*). In the case of ribosomes bearing Met-Lys-tRNA$^{Lys}$ in the A site, we instead observed little to no change in tRNA configurations between the two types of ribosomes (*Figure 4—figure supplement 1*). These findings indicate that pseudouridine modifications at rRNA residues 609 and 863 have the capacity to alter the dynamic properties of pre-translocation ribosome complexes in a way that likely depends on the tRNA species in the P and A sites. Taken together, these smFRET data provide compelling evidence that ribosomes lacking these two specific pseudouridine modifications exhibit functionally relevant physical distinctions in regard to dynamic structural features within the pre-translocation complex, that appear to depend on the mRNA coding sequence.

The observed impacts on aa-tRNA selection and pre-translocation complex dynamics lead us to investigate the sensitivity of HuH-7 sgRNA-ctrl or sgRNA-24 cells to ribosome-targeting drugs (*Figure 4H* and *Figure 4—figure supplement 2*). In doing so, we found that HCC cells lacking *SNORA24* displayed a specific and increased tolerance to Anisomycin (ANS), a drug that binds to the ribosomal A site (*Hansen et al., 2003*) and inhibits peptidyl transfer (*Figure 4H*). This result is in perfect agreement with our smFRET observations that loss of *SNORA24*-guided modifications increase the efficiency of aa-tRNA selection, which would be expected to reduce the time window during each elongation cycle that the ribosome is sensitive to ANS. Collectively, these investigations provide compelling evidence that snoRNA-mediated changes in the chemical composition of mammalian ribosomes, have the potential to affect tRNA selection, ribosome dynamics, and sensitivity of cancer cells to specific translation inhibitors.

## *SNORA24*-guided modifications within the small ribosomal subunit function to enhance translational accuracy

Given that reduced *SNORA24* expression had no observed impact on overall protein production (*Figure 3C and D*), we predicted that the differences in aa-tRNA selection and conformational dynamics observed by smFRET in ribosomes lacking *SNORA24*-guided pseudouridine modifications may impact the accuracy of translation, in a codon-specific manner. To examine to what extent *SNORA24*-guided modifications impact decoding accuracy or translation fidelity in HCC cells, we employed established luciferase reporter systems, consisting of the Renilla luciferase (Rluc) gene fused to the Firefly luciferase (Fluc) gene, to monitor both amino acid misincorporation (*Kramer et al., 2010*) and stop codon readthrough (*Jack et al., 2011*).

We first evaluated decoding accuracy in HCC cells using two luciferase reporters, in which one of two codons in Fluc, either codon 245 (CAC, His) or codon 529 (AAA, Lys), had been mutated to near-cognate codons (CGC and AAU, respectively) (*Figure 5A*, top panel). These mutations are known to reduce Fluc activity (*Kramer et al., 2010*). Misreading of these codons can therefore lead to the incorporation of the original amino acid, to restore Fluc activity, enabling comparative estimates of miscoding error. For both codons tested, we found that HCC cells with reduced *SNORA24* expression display a 10–20% increased level of amino acid misincorporation compared to control cells (*Figure 5A*, bottom panel). We also tested the ability of cells lacking *SNORA24*-guided pseudouridine modifications to terminate translation at stop codons using a similar luciferase reporter system in which a stop codon is placed between Rluc and Fluc (*Figure 5B*, top panel). Interestingly, in the presence of paromomycin, an antibiotic known to induce translation error and stop codon readthrough (*Kramer et al., 2010*), we observe a *SNORA24*-dependent increase (~15%) in UGA stop codon readthrough (*Figure 5B*, bottom panel). By contrast, we detected no difference in UAG stop codon readthrough in the same cells (*Figure 5B*, bottom panel). These findings suggest that Ψ609 and Ψ863 within the small ribosomal subunit function to enhance translational accuracy and that reduced expression and activity of *SNORA24* in HCC may lead to errors in the translation of specific mRNAs.

## Discussion

Ribosome dysfunction and alterations in translation are linked to cancer development (*Pelletier et al., 2018*; *Silvera et al., 2010*; *Sulima et al., 2017*; *Truitt and Ruggero, 2016*). The function of the vast majority of conserved RNA modifications within the ribosome (*Decatur and*

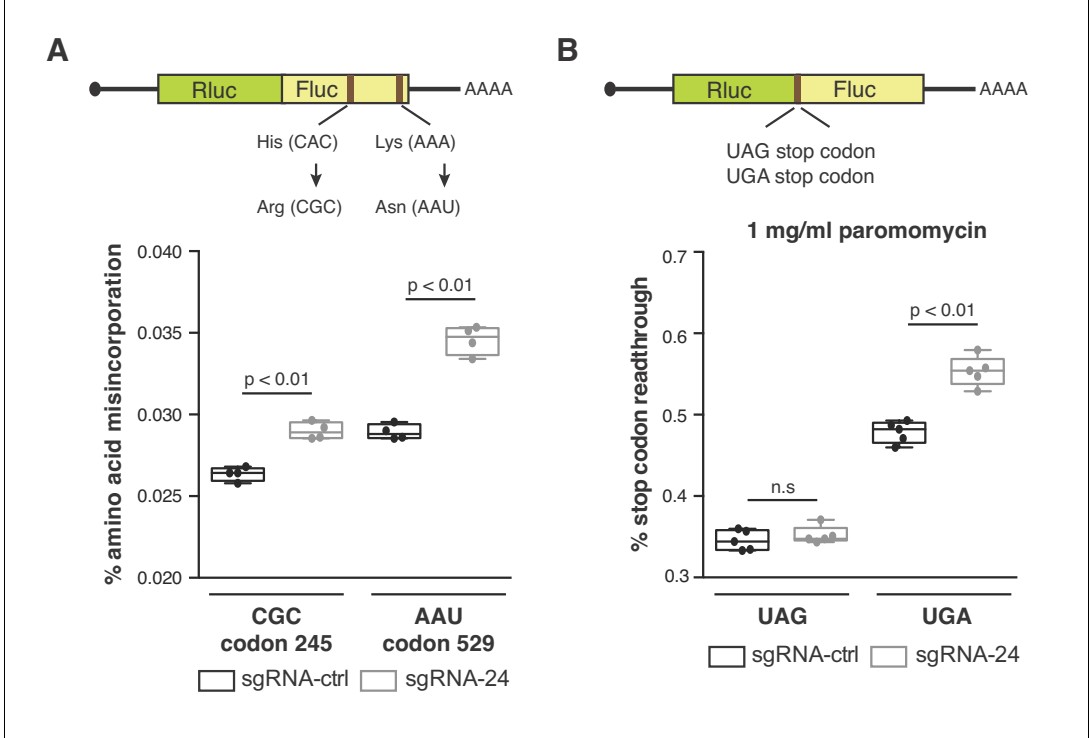

**Figure 5.** *SNORA24*-guided modifications impact translation accuracy. (**A**) Diagram of luciferase reporters used to monitor amino acid misincorporation. Point mutations in Fluc at either His 245 or Lys 529 to near-cognate codons are highlighted (top panel). Amino acid misincorporation (%) of the indicated luciferase reporters (CGC codon or AAU codon) in HuH-7 sgRNA-ctrl cells (black) and HuH-7 sgRNA-24 cells (gray) are shown (bottom panel). On the box and whisker plots, the center line is the medium amino acid misincorporation (%), box limits are minimum and maximum values, and whiskers are S.D. from four independent experiments. Statistical analysis was performed using an unpaired Student's t-test, p<0.01. (**B**) Diagram of luciferase reporters used to monitor stop codon readthrough (top panel). Stop codon readthrough (%) of the indicated luciferase reporters (UAG stop codon or UGA stop codon) in the presence of 1 mg/ml paromomycin in HuH-7 sgRNA-ctrl cells (black) and HuH-7 sgRNA-24 cells (gray) are shown (bottom panel). On the box and whisker plots, the center line is the medium stop codon readthrough (%), box limits are minimum and maximum values, and whiskers are S.D. from five independent experiments. Statistical analysis was performed using an unpaired Student's t-test, p<0.01 and n. s = non significant.

DOI: https://doi.org/10.7554/eLife.48847.020

*Fournier, 2002*), in this context remain largely unexplored. For decades, snoRNAs have been thought to exert minor functions, largely due to findings that in unicellular organisms, depletion of individual snoRNAs yield little to no growth defect. In this respect, the biological influence of snoRNA-directed rRNA modifications in mammalian physiology, and in disease states has been poorly understood (*Williams and Farzaneh, 2012*). We provide evidence that one H/ACA snoRNA plays a direct role in specific steps of cancer development. Our studies reveal that *SNORA24* is implicated in a tumor suppressor program in vivo to halt cellular transformation and with compelling human and mouse data supports a role for *SNORA24* dysfunction in tumor initiation and in maintenance of RAS-driven cancer. Interestingly, in the context of HCC, *SNORA24* appears dispensable for overall protein synthesis, suggesting that cancer-associated changes in *SNORA24* may have more selective functions, for example towards controlling translation of specific mRNAs. This is supported by the apparent codon specificity in aa-tRNA selection and pre-translocation complex dynamics observed by smFRET analysis of ribosomes from cancer cells lacking *SNORA24*-dependent pseudouridine modifications. One intriguing possibility is that *SNORA24*-guided modifications directly impinge on the production of proteins involved in, for example, lipid metabolism and signaling, due to the strong association between *SNORA24* and tumor lipid deposition identified in this study. It is interesting to note a previous link between distinct C/D snoRNAs and response to metabolic stress (*Jinn et al., 2015*; *Michel et al., 2011*). Although the roles of C/D snoRNAs in modulating ribosome activity or translation in this context was not examined, it will be important to determine whether

dysfunction of specific snoRNAs and site-specific ribosome modifications are implicated in human lipid metabolic disorders and/or in the progression of chronic liver disease to HCC.

Our studies employing smFRET imaging represent, to our knowledge, the first report of biophysical changes in ribosomes from cancer cells lacking specific rRNA nucleotide modifications. Ψ609 is located in a highly structured element (the so-called '530 loop' in bacteria), a region of rRNA directly implicated in codon-anticodon pair recognition in the ribosomal A site. Previous studies implicate this region as contributing to the process of domain closure, a large-scale movement within the small subunit that aids mRNA codon-tRNA anticodon pairing required for accurate decoding at the A site (*Fislage et al., 2018*; *Loveland et al., 2016*; *Ogle et al., 2001*; *Ogle et al., 2002*). Our data demonstrating that loss of Ψ609 and Ψ863 enhances the efficiency of aa-tRNA accommodation, imply that *SNORA24*-guided modifications, and Ψ609 in particular, may function to alter domain closure strength. In contrast to Ψ609, Ψ863 is located in an rRNA expansion segment (ES) that is distal to the decoding center of the ribosome. Interestingly, a recent study has established a connection between ribosome expansion segments and translation fidelity (*Fujii et al., 2018*). Thus, it seems plausible that the functional alterations observed in ribosomes lacking Ψ609 and Ψ863 may arise from both direct and indirect mechanisms that alter how the small subunit of the ribosome recognizes and responds to aa-tRNA binding at the A site. In purified ribosomes lacking *SNORA24*-guided modifications, the observed differences in aa-tRNA selection of tRNA$^{Phe}$ compared to tRNA$^{Lys}$ can have several plausible explanations. For instance, studies have shown that the efficiency of aa-tRNA selection varies depending on the tRNA species and the codon in the A site (*Pavlov and Ehrenberg, 2018*; *Zhang et al., 2016*). These and other observations indicate that the stability of intermediates in the decoding process are affected by even small differences in interaction energy between the ribosome and the tRNA substrate. As noted above, decoding intermediates also exhibit conformational changes in the small subunit proximal to the sites of *SNORA24*-guided modifications. We therefore speculate that pseudouridylation of U609 and U863 may fine tune decoding in a manner that results in small, but significant codon-dependent effects on tRNA selection efficiency. In addition, it seems likely that factors such as tRNA abundance, codon-anticodon base pair interactions, and codon context, that are known to influence ribosome elongation rate and translation efficiency (*Gardin et al., 2014*; *Goodarzi et al., 2016*; *Pop et al., 2014*; *Quax et al., 2015*; *Riba et al., 2019*), may lead to gene-specific alterations in translational control in cancer cells with altered *SNORA24* expression.

Our findings that a distinct snoRNA impacts translational accuracy is in line with observations in other organisms, demonstrating that clusters of snoRNA-directed modifications close to the decoding center influence translational fidelity (*Baudin-Baillieu et al., 2009*). However, unlike single-celled organisms that can tolerate relatively high levels of translation errors, minor defects in translation accuracy arising from snoRNA dysfunction in mammals may yield more severe cellular consequences in certain contexts, for example in response to oncogenic stress. For instance, alterations in aa-tRNA selection efficiency or fidelity during translation elongation may lead to changes in the abundance of specific proteins beneficial to cancer cell survival. Differences in translation fidelity may also alter stop codon readthrough during translation termination, as well as the rate at which ribosomes translocate through a given mRNA open reading frame due to impacts on translocation (*Alejo and Blanchard, 2017*). Elucidating the structural-functional roles of modified residues within the ribosome will undoubtedly aid in the development of a new generation of cancer therapeutics focused on targeting ribosomes with differential modification patterns in cancer and will shed significant new light on the role of ancient RNA modifications in directing ribosome activity and cellular integrity.

# Materials and methods

### Key resources table

| Reagent type (species) or resource | Designation | Source or reference | Identifiers | Additional information |
|---|---|---|---|---|
| Genetic reagent (*Mus musculus*) | Wild-type C57BL/6J | The Jackson Laboratory | #000664; RRID: IMSR_JAX:000664 | |

*Continued on next page*

Continued

| Reagent type (species) or resource | Designation | Source or reference | Identifiers | Additional information |
|---|---|---|---|---|
| Genetic reagent (*Mus musculus*) | *Kras*G12D | PMID:15093544 | MGI:2429948 | |
| Genetic reagent (*Mus musculus*) | *Alb*-cre | PMID:9867845 | MGI:2176228 | |
| Cell line (*Homo sapiens*) | Primary human skin fibroblasts | Coriell Cell Repositories | Cat. # GM00730; RRID:CVCL_L944 | |
| Cell line (*Homo sapiens*) | HuH-7 | Japanese Collection of Research Bioresources Cell Bank | Cat. # JCRB0403; RRID:CVCL_0336 | |
| Cell line (*Homo sapiens*) | 293T | American Type Culture Collection | Cat. # CRL-3216; RRID:CVCL_0063 | |
| Cell line (*Mus musculus*) | *Kras*G12D HCC cell line | PMID:30643286 | | Laboratory of Davide Ruggero (UCSF) |
| Antibody | RAS (rabbit polyclonal) | Cell Signaling Technology | Cat. #3965; RRID:AB_2180216 | (1:1000) |
| Antibody | PTEN (rabbit monoclonal) | Cell Signaling Technology | Cat. #9188; RRID:AB_2253290 | (1:1000) |
| Antibody | β-actin (mouse monoclonal) | Sigma-Aldrich | Cat. # A5316; RRID:AB_476743 | (1:10,000) |
| Antibody | p21 (mouse monoclonal) | BD Biosciences | Cat. # 556431 (clone SXM30); RRID:AB_396415 | (1:50) |
| Antibody | NRAS (mouse monoclonal) | Santa Cruz Biotechnology | Cat. # sc-31 (clone F155); RRID:AB_628041 | (1:50) |
| Recombinant DNA reagent | *NRAS*G12V | Addgene, PMID:19147555 | Plasmid #20205; RRID:Addgene_20205 | (pT/Caggs-NRASV12) |
| Recombinant DNA reagent | SB13 | Addgene, PMID:19147555 | Plasmid #20207; RRID:Addgene_20207 | (PT2/C-Luc//PGK-SB13) |
| Recombinant DNA reagent | *HRAS*G12V | Addgene | Plasmid #9051; RRID: Addgene_9051 | (pBABE puro H-Ras V12) |
| Recombinant DNA reagent | *PTEN* shRNA | PMID:29720449 | | (pLKO.1-PTEN-shRNA) Laboratory of Davide Ruggero (UCSF) |
| Recombinant DNA reagent | Rluc-Fluc control | PMID:30576652 | | (pCMV-WT: CMV promoter, Rluc-Fluc) Laboratory of Maria Barna (Stanford University) |
| Recombinant DNA reagent | CGC codon 245 | Other | | (pCMV-245 CGC: CMV promoter, Rluc-Fluc) Laboratory of Maria Barna (Stanford University) |
| Recombinant DNA reagent | AAU codon 529 | This paper | | Generated by site-directed mutagenesis of plasmid pCMV-WT at codon 529 from AAA to AAT (pCMV-529 AAU) |

*Continued*

| Reagent type (species) or resource | Designation | Source or reference | Identifiers | Additional information |
|---|---|---|---|---|
| Recombinant DNA reagent | Readthrough control | PMID:22099312 | | (pJD175f (pHDL-SV40-control)) Laboratory of Jonathan Dinman (University of Maryland) |
| Recombinant DNA reagent | UAG stop codon | Other | | (pJD1644 (pHDL-SV40-UAG)) Laboratory of J onathan Dinman (University of Maryland) |
| Recombinant DNA reagent | UGA stop codon | Other | | (pJD1645 (pHDL-SV40-UGA)) Laboratory of Jonathan Dinman (University of Maryland) |
| Sequence-based reagent | Oligonucleotides for qPCR analysis | This paper | | *Supplementary file 2* |
| Sequence-based reagent | Synthetic mRNA Met-Phe | Dharmacon | | CAACCUAAAACU UACACACCCUUAGAGGGAC AAUCGAUGUUCAAAGUC UUCAAAGUCAUC |
| Sequence-based reagent | Synthetic mRNA Met-Lys | Dharmacon | | CAACCUAAAACUUAC ACACCCUUAGAGGGACAAUC GAUGAAAUUCGU CUUCAAAGUCAUC |
| Commercial assay or kit | Dual-Luciferase Reporter Assay System | Promega | Cat. # E1910 | |
| Commercial assay or kit | Senescence Detection Kit | Calbiochem-Millipore | Cat. # QIA117 | |
| Commercial assay or kit | CellTiter-Glo Luminescent Cell Viability Assay | Promega | Cat. # G7570 | |
| Chemical compound, drug | Anisomycin | Sigma-Aldrich | Cat. # A9789 | |
| Chemical compound, drug | Paromomycin | Sigma-Aldrich | Cat. # P9297 | |
| Chemical compound, drug | Cycloheximide | Sigma-Aldrich | Cat. # C7698 | |
| Chemical compound, drug | O-propargyl-puromycin | Medchem Source LLP | Cat. # JA-1024 | |
| Chemical compound, drug | Cy3-Maleimide | GE Healthcare | Cat. # PA23031 | |
| Chemical compound, drug | LD655-NHS | Lumidyne Technologies | Cat. # 08 | |
| Chemical compound, drug | LD650-NHS | Lumidyne Technologies | Cat. # 99 | |
| Software, algorithm | PyMOL | Schrödinger, NY, USA | https://www.pymol.org/2/ | |
| Software, algorithm | ImageJ | National Institute of Health, USA | https://imagej.nih.gov/ij/ | |
| Software, algorithm | GraphPad Prism six software | GraphPad | https://www.graphpad.com/ | |
| Software, algorithm | Spartan | Other | | Available at: https://www.scottcblanchardlab.com/software |

## Animal studies

Expression of $NRAS^{G12V}$ in mouse liver was performed as previously described (*Kang et al., 2011*), with minor modifications. Briefly, C57BL/6 wild-type mice 8–12 weeks old were injected with a 5:1 molar ratio of transposon ($NRAS^{G12V}$) to transposase (SB13) encoding plasmids (35 µg total DNA) by hydrodynamic tail vein injection. As a control, mice were injected with the transposase (SB13) encoding plasmid (35 µg total DNA) by hydrodynamic tail vein injection (SB(-)$NRAS^{G12V}$). Plasmid DNA were prepared using a Qiagen Endo Free Maxi Kit. DNA was suspended in Normal Saline solution and administrated at a final volume of 10% of the animal's body weight. Mice treated with locked nucleic acid (LNA-ctrl or LNA-24) (Exiqon, MA, USA) were tail vein injected with 20 mg/kg LNA three days prior to hydrodynamic delivery of $NRAS^{G12V}$ (SB(+)$NRAS^{G12V}$) or control (SB(-)$NRAS^{G12V}$) and received LNA treatment every 10 days for the duration of the study. The LNA sequences were as follows: LNA-ctr 5'-AACACGTCTATACGC-3' and LNA-24 5'-GCTCTTCCATGGCTAG-3'. For determination of senescence, mouse livers were harvested 6 days after $NRAS^{G12V}$ administration. Orthotopic injections of *Alb*-cre;$Kras^{G12D}$ mice liver tumor cells into the subcapsular region of the median liver lobe of C57BL/6 wild-type mice were performed as previously described (*Xu et al., 2019*). All mice were maintained under specific pathogen-free conditions. Experiments were performed in compliance with guidelines approved by the Institutional Animal Care and Use Committee (IACUC) with assistance from the Laboratory Animal Resource Center (LARC) of UCSF.

## Cell culture and reagents

Primary human skin fibroblast (GM00730) were obtained from Coriell Cell Repositories (Coriell Institution for Medical Research, NJ, USA) and maintained in (Dulbecco's Modified Eagle Medium (DMEM) supplemented with 10% Fetal Bovine Serum and Penicillin/Streptomycin (DMEM, 10% FBS, P/S). HuH-7 are an established cell line obtained from the Japanese Collection of Research Bioresources Cell Bank (JCRB0403) of the National Institutes of Biomedical Innovation, Health and Nutrition, Japan and maintained in DMEM, 10% FBS, P/S. Generation of mouse liver cancer cell lines (from *Alb*-cre;$Kras^{G12D}$ mice) was previously described (*Xu et al., 2019*) and maintained in DMEM, 10% FBS, P/S. 293 T cells were obtained from ATCC and maintained in DMEM, 10% FBS, P/S. All cell lines used in this study were found to be negative of mycoplasma contamination using a MycoAlert mycoplasma detection kit (Lonza, Allendale, NJ, USA). Retroviral and lentiviral particles were produced in 293 T cells by transfection with the appropriate expression and packaging plasmids using PolyFect Transfection Reagent (Qiagen) and filtering cultured supernatants through a 0.45 µM filter. Early passage primary skin fibroblasts (P9) were infected with *PTEN* shRNA or $HRAS^{G12V}$ expression constructs followed by selection with puromycin (2 µg/ml). Retroviral vectors were obtained from Addgene (pBabe puro $HRAS^{G12V}$ (#9051)). Lentiviral vector harboring a shRNA targeting *PTEN* (pLKO.1 backbone) was previously described (*Nguyen et al., 2018*). All chemicals used in this study were purchased from Sigma-Aldrich unless otherwise stated.

## Gene editing using Cas9-guide RNA ribonucleoprotein (RNP) complexes

All sgRNAs targeting mouse or human *SNORA24* were designed using the Zhang Lab design tool (crispr.mit.edu). Chemically modified synthetic sgRNAs were purchased from Synthego (Menlo Park, CA, USA) and Cas9-NLS purified protein was from the QB3 MacroLab (UC Berkeley, CA, USA). Cas9 RNP was prepared immediately prior to nucleofection by incubating Cas9 protein with sgRNA at 1:1.3 molar ratio in 20 mM HEPES (pH 7.5), 150 mM KCl, 1 mM $MgCl_2$, 10% glycerol and 1 mM TCEP at 37°C for 10 min. Cells were dissociated using trypsin, pelleted by centrifugation, and washed once with D-PBS. Nucleofection of human HuH-7 cells and mouse $Kras^{G12D}$ tumor cell line was performed using Amaxa Cell Line Nucleofector Kit V (Lonza, Allendale, NJ, USA) and program H-022 on an Amaxa Nucleofector II system. Each nucleofection reaction consisted of ~$4\times10^5$ cells in 50 µl of nucleofection reagent mixed with two distinct 10 µl RNP mixtures containing different sgRNA (to allow specific deletion within the *SNORA24* gene locus [sgRNA-24]). Cas9 alone or a set of non-targeting control sgRNA (sgRNA-ctrl) were used in a separate RNP reaction. Two days following nucleofection, gene editing was confirmed by extracting genomic DNA (gDNA) from cells using Quick Extraction (Lucigen Corporation, WI, USA), performing PCR of the *SNORA24* loci using gene specific primer, and sequencing the PCR product. The following sgRNA sequences were use; *SNORA24* human sgRNA #1 5'-GGATATGCTCTTCCATGGCT-3', *SNORA24* human sgRNA #2 5'-

CAAAGCTGTCACCATTTAAT-3', non-targeting sgRNA #1 5'-AACGACTAGTTAGGCGTGTA-3', non-targeting sgRNA #2 5'-CGCCAAACGTGCCCTGACGG-3', *Snora24* mouse sgRNA #1 5'-TCTTTGGGACCTGCCGCCTG-3', *Snora24* mouse sgRNA #2 5'-CACTTGCTCAAGTCAGAATC-3'.

## Polysome fractionation

HuH-7 sgRNA-ctrl and sgRNA-24 cells were incubated with 100 µg/ml cycloheximide (Sigma) in the growth media for 5 min at 37°C and 5% $CO_2$. Cells were washed once in ice-cold PBS containing 100 µg/ml cycloheximide. Cells were then scraped in 5 ml of ice-cold PBS containing 100 µg/ml cycloheximide and pelleted. Cell pellets were lysed in buffer containing 10 mM Tris-HCl (pH 8), 150 mM NaCl, 1.5 mM MgCl2, 1% Triton X-100, 20 mM DTT, 150 µg/ml cycloheximide, and 640 U/ml Rnasin for 30 min on ice. Lysates were centrifuged at 10,000 x g for 5 min at 4°C. The supernatant (~300 ul) were adjusted by OD260 (to OD260 of ~15) and loaded onto a 10–50% sucrose gradient before centrifugation at 37,500 rpm for 2.5 hrs at 4°C in a Beckman L8-70M ultracentrifuge. Samples were separated on an ISCO gradient fractionation system to evaluate polysome profiles.

## Western blot analysis

Western blot analysis was performed on samples lysed in RIPA buffer (50 mM Tris pH 8, 150 mM NaCl, 0.2% Na deoxycholate, 0.5% TritonX-100) with the addition of PhosSTOP and Complete Mini proteasome inhibitors (Roche) using standard procedures with commercial antibodies for RAS (Cell Signaling Technology), PTEN (Cell Signaling Technology) and β-actin (Sigma).

## Immunohistochemistry

Immunohistochemistry analysis was performed on OCT embedded frozen tissue using standard protocols and the following primary antibodies: NRAS (Santa Cruz Biotechnology) and p21 (BD Biosciences).

## Senescence determination

Cellular senescence was assayed 15 days after retroviral expression of *HRAS*[G12V] in primary human skin fibroblasts using a senescence detection kit (Calbiochem) according to manufacturer's instructions. Cells were imaged using a Nikon TE2000E inverted microscope. Determination of senescence in liver sections or whole liver lobes was carried out as described previously (*Kang et al., 2011*).

## ORO staining and quantification in human HCC specimens and mouse liver tissue

OCT embedded frozen tissue was prepared using standard protocols and following equilibrated at room temperature for 10 min, sections were fixed in formalin for 5 min. Following wash in tap water, slides were stained in ORO working solution for 10 min at room temperature. ORO stock and working solution were prepared as previously described (*Mehlem et al., 2013*). Slides were washed in tap water for 10 min and counterstained with Mayer's hematoxylin by submerging the slides in hematoxylin for 3 min. Slides were rinsed under running tap water for ~10 min and mounted with AquaSlip from AMTS Inc (Lodi, CA, USA). Hematoxylin and Eosin (H and E) stained frozen sections were imaged on an Aperio Versa slide scanner (Leica Biosystems), equipped with a HC PL Fluotar 10X/.32 objective. ORO stained frozen sections were imaged on an Axio Scan.Z1 slide scanner (Carl Zeiss Microscopy), equipped with a Plan-Apochromat 10x/0.45 objective. ORO quantification was performed by selecting four regions of interest (ROIs) from each H and E. scn image file and extracted as a TIFF at 1000 × 1000 pxl (551 × 551 microns; scale = 551 nm/pxl) in Aperio ImageScope v.12.3.2 software. The analogous region from the ORO stained serial adjacent section. czi image file (scale = 442 nm/pxl) was extracted in Zen, converted to TIFF and downsampled to yield the same spatial scaling for analysis. The two resulting TIFF files were spatially registered using ImageJ TrackEM2. An ORO RGB image was used as input to tune the threshold parameters in the Zen Pro Image Analysis software module for quantifying lipid droplets. Three classes were created; ORO (Oil Red O positive), All (all tissue), and White (empty) with the following colormetric parameters: ORO: R 94–158, G 2–98, B – 109; All: R 90, G 10–177, B 19–182; White: R 191–220, G 177–210, B 181–224. The percentage (%) ORO positive stained area per total area examined from four different ROIs of each tissue section was calculated and summed. For each condition tested, the

mean ± SD ORO positive area (%) was plotted, with the Y-axis label representing 'ORO positive area (%)". For mouse tissue sections (n = 3 mice), ROIs were identified within the H and E images that could be classified as abnormal or normal tissue under the guidance of a pathologist. Abnormal or normal tissue ROIs (fixed area of 7400 pixels) were applied to the analogous region within the registered ORO image (as shown in *Figure 2—figure supplement 1*). For human HCC specimens, patient samples were dichotomized into high or low *SNORA24* expression by identifying samples with *SNORA24* expression more extreme than ±1 SD from the mean (n = 17 HCC specimens) and ORO staining and quantification was performed as described above.

## Measurement of protein synthesis by OPP incorporation

OP-Puro (Medchem Source LLP, WA, USA) was reconstituted in PBS, adjusted to pH 6.5, and stored in aliquots at −20˚C. Cells were treated with 30 µM OP-Puro or PBS (mock to subtract background signal during analysis). Two hours following OP-Puro addition to the media, cells were dissociated using trypsin, pelleted by centrifugation, washed in PBS, and fixed in paraformaldehyde (PFA) in PBS for 15 min on ice. After washing in cold PBS, samples were permeabilized in the dark using PBS with 3% FBS and 0.1% saponin. Click-iT reaction (Invitrogen) was performed according to manufacturer's instructions with cycloaddition conjugation to Alexa555 for 30 min at room temperature with light protection. Data was acquired using a BD LSRII and analyzed with FlowJo to calculate the fluorescence intensity of each sample. For quantification, the relative rates of protein synthesis depicted by OP-Puro signals were calculated as mean fluorescence intensity (MFI), subtracting the auto-fluorescence background from mock (PBS control). Normalized MFI for each cell sample was plotted with SD of the mean.

## Patient samples

Liver tissue specimens were obtained from patients undergoing treatment for HCC at the University of California, San Francisco (San Francisco, CA, USA). A summary of patient demographics and staging is presented in *Supplementary file 1*.

## Microarray gene expression data analysis for H/ACA snoRNAs

Microarray gene expression was obtained from the NCBI GEO database, accession GSE25097 (HCC), accession GSE22898 (Diffuse large B-cell lymphoma), accession GSE20916 (Colorectal cancer), accession GSE28735 (Pancreatic ductal adenocarcinoma), and accession GSE43458 (Lung adenocarcinoma). Expression data from probes corresponding to H/ACA snoRNAs were extracted, analyzed for fold change in expression in tumor and control samples, and statistical significance was calculated using paired or unpaired Student's t-test as indicated.

## HCC patient survival analysis

Microarray gene expression and clinical data were obtained from the NCBI GEO database, accession GSE25097 (*Hao et al., 2011*; *Kan et al., 2013*). Calculations were performed in R (*R Development Core Team, 2008*). Patient samples were dichotomized into high or low by identifying samples with *SNORA24* expression more extreme than ±one SD from the mean (n = 24). Kaplan-Meier survival curves were fit, and statistical significance was calculated using the log-rank test, with p<0.05 used as a threshold for statistical significance. Similar analyses were performed for *SNORA14B, SNORA17, SNORA67, SNORA72*, and *SNORA81*.

## Single-molecule FRET microscopy

All smFRET experiments were conducted at 37˚ C in human polymix buffer (50 mM Tris pH 7.5, 5 mM MgCl$_2$, 50 mM NH$_4$Cl, 2 mM spermidine, 5 mM putrescine) containing a mixture of triplet-state quenchers (1 mM Trolox, 1 mM *4*-nitrobenzyl alcohol (NBA), 1 mM cyclooctatetraene (COT)) and an enzymatic oxygen scavenging system (2 µM 3,4-Dihydroxybenzoic acid (PCA), 0.02 Units/ml protocatechuate *3,4*-dioxygenase (PCD)). Ribosomes from HuH-7 sgRNA-ctrl or sgRNA-24 cells were prepared using the protocol described in *Flis et al. (2018)*. Elongation factor eEF1A and fluorescence labeled tRNAs were prepared as in *Flis et al. (2018)*. Pre-formed 80S initiation complexes made with ribosomes from either HuH-7 sgRNA-ctrl or sgRNA-24 cells, containing Cy3-labeled Met-tRNA$^{f-Met}$ in the P site and displaying either a UUC or AAA codon in the A site, were surface-immobilized

on passivated quartz coverslips (*Blanchard et al., 2004*) in a home-built total internal reflection-based fluorescence microscope (*Juette et al., 2016*). To initiate tRNA selection, 10 nM ternary complex, consisting of eEF1A, GTP and either Phe-tRNA$^{Phe}$ labeled with LD655 or Lys-tRNA$^{Lys}$ labeled with LD650 was stop flow delivered to the immobilized ribosomes. Imaging of the pre-translocation complexes was carried out by washing ternary complex from the flow cell with polymix buffer 30 s after injection. smFRET data were recorded at a time resolution of 15 ms at ~0.25 kW/cm$^2$ laser (532 nm) illumination. Donor and acceptor fluorescence intensities were extracted from the recorded movies and FRET efficiency traces were calculated. FRET traces were selected for further analysis according to the following criteria: a single catastrophic photobleaching event, at least 8:1 signal/background-noise ratio and 6:1 signal/signal-noise ratio, less than four donor-fluorophore blinking events and a correlation coefficient between donor and acceptor <0.5.

smFRET traces were analyzed using hidden Markov model idealization methods as implemented in the SPARTAN software package (*Juette et al., 2016*). The idealization model for tRNA selection traces included four separate FRET values accounting for unbound, initial binding, GTPase activation, and accommodated states during tRNA selection (*Ferguson et al., 2015*; *Geggier et al., 2010*) with FRET values of 0.0 ± 0.05, 0.2 ± 0.075, 0.46 ± 0.075 and 0.72 ± 0.075. The idealization model for the pre-translocation state included three FRET states with FRET values of 0.22 ± 0.075, 0.42 ± 0.075 and 0.72 ± 0.075 (0.61 ± 0.05 in case of LD650 labeled tRNA$^{Lys}$ containing ribosomes) accounting for the hybrid 1, hybrid two and classical tRNA binding states (*Pellegrino et al., 2019*). To generate cumulative distributions for estimation of apparent reaction rates during tRNA selection the number of traces that had achieved the 0.72 FRET state prior to each movie frame were summed. An exponential function containing two exponential terms and a term accounting for the initial delay due to the stop flow delivery dead time was then fit to the data. All distributions contained two phases, a fast phase accounting for >85% of events and a slower phase accounting for the remainder. In all cases the reaction rate of the dominant, fast, phase was used for further analysis. To take the effect of donor photobleaching into account for estimation of accurate tRNA selection $k_{cat}/K_M$ values, donor photobleaching rates estimated from the total dataset were subtracted from the apparent reaction rates. To estimate the fraction of time the ribosomal pre-translocation complexes spend in each tRNA binding state, state dwell times were extracted directly from the hidden Markov-model idealizations. All experimental uncertainties were estimated from bootstrap analysis of two to five smFRET datasets. Significance testing of the difference in tRNA selection efficiency and tRNA state occupancy between ribosomes isolated from HuH-7 sgRNA-ctrl and sgRNA-24 cells was carried out by a bootstrap implementation of Welch's t-test. Briefly, the t statistic was calculated from the bootstrap distributions of the estimated reaction rate constants or fractional state occupancies. This was then compared to $10^6$ t statistics calculated from bootstrap samples picked from null distributions generated by shifting the mean of both original distributions to their pooled mean. This generated the upper bound for the *P* value of p<10$^{-6}$ reported in the text.

## LipidTOX staining

HuH-7 sgRNA-ctrl or sgRNA-24 cells plated on glass coverslips and treated with Oleic Acid (diluted 1:10 in media) 24 hrs after plating. 6 hrs following Oleic Acid addition, cells were fixed using 4% PFA for 30 min at room temperature, followed by a PBS wash, and LipidTOX green neutral lipid staining (Thermo) (1:200 dilution in PBS) for 1 hr. Coverslips were mounted on glass slides using Prolong anti-fade mounting solution with DAPI. Imaging was performed on a Zeiss Cell Observer Spinning Disc Confocal Microscope and quantification was performed using ImageJ.

## Proliferation assay

HuH-7 sgRNA-ctrl or sgRNA-24 cells were plated at 2,000 cells per well in 96-well plates. 24 hrs after plating, cells were treated with the indicated concentration of translation inhibitor or DMSO and incubated for 48 hrs. CellTiter-Glo Luminescent Cell Viability Assay (Promega, WI, USA) was performed following manufacturer's instructions with luminescence measurements made using a Glomax 96-well plate luminometer (Promega). Proliferation data were generated by first normalizing luminescence intensity in each well to that of the DMSO-treated wells and normalized luminescence data was plotted (± SD) from at least three independent experiments.

## Quantification of site-specific rRNA pseudouridine modifications

SCARLET was performed essentially as previously described (*Li et al., 2015*; *Liu and Pan, 2015*) on 10 µg of total RNA isolated using TRIzol (Invitrogen) from the indicated human HuH-7 cells or mouse *Kras*<sup>G12D</sup> tumor cell lines using the following oligonucleotides, where Nm = 2'-O-Me modified nucleotide; U609 18S rRNA chimera: 5'-CmAmGACTUmGmCmCmCmUmCmCmAmAmUm-3', U609 18S rRNA splint: 5'-AGCTGGAATTACCGCGGCTGCTGGCACCACTATTAACTCACAGGACCGGCGA TGGCTG-3', U863 18S rRNA chimera: 5'-UmCmCmAmUmUmAmTTCCUmAmGmCmUmGmCm-3', U863 18S rRNA splint: 5'-CAAAATAGAACCGCGGTCCTATTCCATTACTATTAACTCACAG-GACCGGCGATGGCTG-3'. Site-specific detection of pseudouridine modifications in LNA-S and LNA-24 treated samples was performed 48 hrs post-transfection as described (*Karijolich et al., 2010*) using the following oligonucleotides, where Nm = 2'-O-Me modified nucleotide; U609 18S rRNA: 5'-CmAmGACTUmGmCmCmCmUmCmCmAmAmUm-3', U863 18S rRNA: 5'-UmAmTTCC UmAmGmCmUmGmCmGmGmUmAm-3', U1731 28S rRNA: 5'-CmAmTTCGCmUmUmUmAmCmC mGmGmAmUm-3', U105 18S rRNA: 5'-GmAmTTTAAmUmGmAmGmCmCmAmUmUmCm-3'. Results were visualized by a phosphor imager and quantification of uridine or pseudouridine in a given sample was determined using ImageJ.

## Luciferase reporter assays

HuH-7 sgRNA-ctrl and HuH-7 sgRNA-24 cells were seeded in 12 well plates at 30,000 cells/well. 24 hrs later cells were transfected using Lipofectamine 2000 (Invitrogen) with 0.1 µg per well of the indicated luciferase reporter construct. Cells were lysed after 24 hrs in passive lysis buffer and Rluc and Fluc activity was assessed using the Dual-luciferase Reporter Assay System (Promega) according to the manufacturer's instructions and using a Glomax microplate luminometer (Promega). For stop codon readthrough experiments, performed in the presence of paromomycin, 1 mg/ml paromomycin (Sigma) was added to cells 8 hrs post-transfection. To measure stop codon readthrough (%), normalized Fluc activity (Fluc/Rluc) from UAG or UGA stop codon readthrough luciferase reporters was further normalized to a control construct, which does not have a stop codon between Rluc and Fluc as described (*Jack et al., 2011*). To measure amino acid misincorporation (%), normalized Fluc activity (Fluc/Rluc) from CGC or AAU amino acid misincorporation luciferase reporters was normalized to a control construct, which does not contain a point mutation in Fluc as described (*Fujii et al., 2018*). The amino acid misincorporation (%) or stop codon readthrough (%) values from the indicated number of independent experiments in HuH-7 sgRNA-ctrl and sgRNA-24 cells are shown.

## Quantitative Polymerase Chain Reaction (qPCR) and snoRNA qPCR array

RNA was isolated using TRIzol (Invitrogen) purification on Direct-zol RNA Microprep columns (Zymo Research, CA, USA) according to manufacturer's instructions with DNase treatment. cDNA samples were diluted 1:10 and 1 µl of template was used in a PowerUP SYBR Green master mix reaction run on an Applied Biosystems QuantStudio 6 Flex Real-Time PCR System (Thermo Fisher). qPCR primer sequences are listed in *Supplementary file 2*. For snoRNA qPCR array, 2 µg Dnase treated (Turbo DNAse) RNA was reverse transcribed using an Arraystar rtStar First-strand cDNA Synthesis kit. The Arraystar nrStar snoRNA PCR Array was performed following manufacturer's instructions using Arraystar SYBR Green Real-time qPCR master mix and run on an Applied Biosystems QuantStudio 6 Flex Real-Time PCR System (Thermo Fisher).

## Quantification and statistical analysis

Unless otherwise stated data is presented as mean ± SD. Statistical tests and specific *P* values used for experiments are listed in the figure legends and were generated using GraphPad Prism six software unless otherwise stated. Results are representative of at least three independent experiments. For survival analysis, a log-rank test was used. p<0.05 was considered significant and the exact *P* values are indicated in the figures and the corresponding figure legends.

## Acknowledgements

We thank members of the Ruggero Laboratory for helpful discussion and Y Xu for technical support. We are grateful to K Fujii and M Barna (Stanford University) for providing luciferase reporters to measure amino acid misincorporation in mammalian cells and to J Dinman (University of Maryland) for providing luciferase reporters to measure stop codon readthrough in mammalian cells. We thank the UCSF Helen Diller Cancer Center Preclinical Therapeutics Core, particularly D Wang. This work was supported by grants from NIH R21 TR001743 (BC), K01 ES028047 (BC), R01 GM079238-13 (SCB) and R35 CA242986 (DR).

## Additional information

### Competing interests

Adrian Contreras: Current employee of Celgene Corporation. John M Luk: Current employee of Arbele Corporation. The other authors declare that no competing interests exist.

### Funding

| Funder | Grant reference number | Author |
|---|---|---|
| National Institutes of Health | R35 CA242986 | Davide Ruggero |
| National Institutes of Health | R01 GM079238-13 | Scott C Blanchard |
| National Institutes of Health | R21 TR001743 | Bin Chen |
| National Institutes of Health | K01 ES028047 | Bin Chen |

The funders had no role in study design, data collection and interpretation, or the decision to submit the work for publication.

### Author contributions

Mary McMahon, Conceptualization, Data curation, Formal analysis, Validation, Investigation, Visualization, Methodology, Writing—original draft, Writing—review and editing; Adrian Contreras, Conceptualization, Formal analysis, Validation, Investigation, Methodology, Writing—review and editing; Mikael Holm, Formal analysis, Validation, Investigation, Visualization, Methodology, Writing—review and editing; Tamayo Uechi, Craig M Forester, Xiaming Pang, Cody Jackson, Meredith E Calvert, Formal analysis, Validation, Investigation, Writing—review and editing; Bin Chen, David A Quigley, Formal analysis, Writing—review and editing; John M Luk, Resources, Writing—review and editing; R Kate Kelley, John D Gordan, Ryan M Gill, Resources, Investigation, Writing—review and editing; Scott C Blanchard, Resources, Supervision, Funding acquisition, Visualization, Methodology, Writing—review and editing; Davide Ruggero, Conceptualization, Resources, Supervision, Funding acquisition, Project administration, Writing—review and editing

### Author ORCIDs

Mary McMahon https://orcid.org/0000-0001-5548-2949
Davide Ruggero https://orcid.org/0000-0002-9444-5865

### Ethics

Human subjects: This study was approved by the Institutional Review Board (IRB) of the University of California, San Francisco (UCSF). Written informed consent was obtained from every patient. Liver tissue specimens were obtained from patients undergoing treatment for HCC at UCSF.

Animal experimentation: This study was performed in accordance with the recommendations in the Guide for the Care and Use of Laboratory Animals of the National Institutes of Health. All of the animals were handled according to approved Institutional Animal Care and Use Committee (IACUC) protocols (AN151649) of the University of California, San Francisco, with assistance from the Laboratory Animal Resource Center (LARC).

**Decision letter and Author response**
Decision letter https://doi.org/10.7554/eLife.48847.025
Author response https://doi.org/10.7554/eLife.48847.026

## Additional files

### Supplementary files
• Supplementary file 1. Summary of HCC patients demographics and staging.
DOI: https://doi.org/10.7554/eLife.48847.021

• Supplementary file 2. Sequences of primers used.
DOI: https://doi.org/10.7554/eLife.48847.022

• Transparent reporting form
DOI: https://doi.org/10.7554/eLife.48847.023

### Data availability
All data generated or analyzed during this study are included in the manuscript and supporting files. Source data files have been provided for Figure 1 and Figure 1—figure supplement 2.

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
