## [Decision Letter]

Thank you for submitting your article "A single H/ACA small nucleolar RNA mediates tumor suppression downstream of oncogenic RAS" for consideration by *eLife*. Your article has been reviewed by two peer reviewers, and the evaluation has been overseen by a Reviewing Editor and James Manley as the Senior Editor. The following individual involved in review of your submission has agreed to reveal their identity: Sohail Tavazoie (Reviewer #2).

The reviewers have discussed the reviews with one another and the Reviewing Editor has drafted this decision to help you prepare a revised submission.

Summary:

The authors document for the first time the role of H/ACA box snoRNA 24 as a tumor suppressor involved in Ras-mediated senescence. SNORA24 mediated modifications are shown to be involved in ribosome dynamics during translation. Consistently, cells lacking SNORA24 are differentially affected by the drug anisomycin, which acts at the same ribosomal region that contains the modification.

Essential revisions:

1) The authors conclude that SNORA24 tumor suppressive activity could be mediated by its function in uridine modification and then on ribosomal function. However, they do not show that reduced modification of SNORA24 targeted uridines is involved during oncogenic transformation. Analysis of site-specific pseudouridylation of U609 and U863 is performed only in cellular clones after CRISPR/CAS9 mediated disruption of SNORA24. In this setting, it is not surprising that pseudouridylation of these sites is affected (since cells completely lack the guide snoRNA). To support the authors' conclusions they should find a reduction in site-specific modification, for instance, in mice liver bypassing OIS after LNA administration or in human tumor specimens characterized by low SNORA24 levels.

2) Snora24 lies within the intron of SNHG8 non-coding RNA (small nucleolar RNA host gene 8). The authors should assess whether LNA treatment impacts the expression of SNHG8.

3) For Figure 2E, the authors should provide sequence validation of CRISPR KO or expression level validation of repressed snoRNA levels.

4) In Figure 1A, the authors should label the primary snoRNAs that they have validated by qPCR to be significantly altered.

5) In the Discussion, the authors propose that SNORA24 could have an effect on ribosomal fidelity both in term of codon usage and stop codon read-through. This could be readily experimentally tested on the cells they have generated by the use of reporter encoding transcripts containing inactivating missense mutations or a premature stop codon as reported in Bellodi et al. Molecular Cell, 2011.

---

## [Author Response]

Essential revisions:1) The authors conclude that SNORA24 tumor suppressive activity could be mediated by its function in uridine modification and then on ribosomal function. However, they do not show that reduced modification of SNORA24 targeted uridines is involved during oncogenic transformation. Analysis of site-specific pseudouridylation of U609 and U863 is performed only in cellular clones after CRISPR/CAS9 mediated disruption of SNORA24. In this setting, it is not surprising that pseudouridylation of these sites is affected (since cells completely lack the guide snoRNA). To support the authors' conclusions they should find a reduction in site-specific modification, for instance, in mice liver bypassing OIS after LNA administration or in human tumor specimens characterized by low SNORA24 levels.

We thank the reviewer for suggesting this experiment. Unfortunately, due to the large amount of starting material required for experiments to detect site-specific rRNA modifications, we were unable to evaluate SNORA24-directed pseudouridine modifications in primary human HCC specimens with low SNORA24 levels due to limited availability of these valuable patient samples.

However, we provide evidence of the specificity of SNORA24 reduction using LNA to decrease SNORA24-guided modifications on 18S rRNA at U609 and U863 (presented in new Figure 1—figure supplement 5D), without impacting pseudouridine modifications at other sites on 18S rRNA and 28S rRNA, that are not guided by SNORA24 (presented in new Figure 1—figure supplement 5E).

2) Snora24 lies within the intron of SNHG8 non-coding RNA (small nucleolar RNA host gene 8). The authors should assess whether LNA treatment impacts the expression of SNHG8.

In the revised version of the manuscript, we provide new evidence that LNA targeting SNORA24 or CRISPR-Cas9 gene editing of the SNORA24 loci have no observed effects on the expression of the SNORA24 host gene, SNHG8. Upon treatment of mice with LNA targeting SNORA24, we observe a significant reduction in SNORA24 expression in liver tissue isolated from at least n=4 mice (as shown in Figure 1—figure supplement 5B). In contrast, in the same tissue we observe little to no detectable difference in SNHG8 levels from at least n=4 independent mice compared to tissue from mice treated with a non-targeting LNA control sequence (presented in new Figure 1—figure supplement 5C). We also present new data demonstrating that CRISPR-Cas9 mediated reduction of SNORA24 in mouse liver cancer cells has no observed influence on the expression of the SNORA24 host gene, SNHG8, compared to controls (presented in new Figure 2—figure supplement 2B).

Lastly, we present new findings that in human primary HCC specimens, in which we observe a statistically significant reduction in SNORA24 in matched tumor versus non-tumor tissue (as shown in Figure 1—figure supplement 4A), we do not detect any change in the expression of the corresponding host gene, SNHG8 (presented in new Figure 1—figure supplement 4B). These findings reveal that the observed SNORA24 deregulation is independent of the expression of the corresponding host gene.

3) For Figure 2E, the authors should provide sequence validation of CRISPR KO or expression level validation of repressed snoRNA levels.

We now provide sequence information for the genomic region targeted by CRISPR-Cas9 gene editing of the SNORA24 loci, using two distinct sgRNAs, that generate an 81 nucleotide deletion in KRAS^G12D^ sgRNA-24 HCC cells (presented in new Figure 2—figure supplement 2A).

In addition, we show the expression levels of SNORA24 in cells used in Figure 2E (left panel), in which we clearly observe reduced SNORA24 levels upon targeting SNORA24 using CRISPR-Cas9 gene editing (Figure 2E, right panel). We further present new data that this reduction in SNORA24 decreases pseudouridine modifications at the expected target residues on 18S rRNA (presented in new Figure 2—figure supplement 2C).

4) In Figure 1A, the authors should label the primary snoRNAs that they have validated by qPCR to be significantly altered.

As suggested by the reviewer, we have now updated Figure 1A to highlight and label the snoRNAs found to exhibit significant alterations in expression (red) that were subsequently validated by qPCR to be significantly altered (as shown in Figure 1—figure supplement 1B).

5) In the Discussion, the authors propose that SNORA24 could have an effect on ribosomal fidelity both in term of codon usage and stop codon read-through. This could be readily experimentally tested on the cells they have generated by the use of reporter encoding transcripts containing inactivating missense mutations or a premature stop codon as reported in Bellodi et al. Molecular Cell, 2011.

We thank the reviewer for suggesting these important experiments. As requested, we have now assessed the effects of SNORA24 on translation fidelity using previously established luciferase reporter systems consisting of the *Renilla* luciferase (Rluc) gene fused to the Firefly luciferase (Fluc) gene and evaluated both amino acid misincorporation and stop codon readthrough.

We evaluated decoding accuracy in HCC cells using luciferase reporters, in which a mutation in Fluc, at either codon 245 (Histidine (His)) or codon 529 (Lysine (Lys) to a near-cognate codon, reduces Fluc activity. Misincorporation of aa-tRNAs at these specific codons increases Fluc activity and therefore enables a readout of translational error (Kramer et al., 2010). Importantly, HCC cells lacking SNORA24 reproducibility exhibit a ~10-15% increase in amino acid misincorporation at these codons (presented in new Figure 5A).

We also tested the ability of cells lacking SNORA24-directed rRNA modifications to terminate translation at stop codons using a similar luciferase reporter system in which a stop codon is placed between Rluc and Fluc (Jack et al., 2011). Interestingly, in the presence of paromomycin, an antibiotic known to induce translation error and stop codon readthrough (Kramer et al., 2010), we observe a significant increase in readthrough of the UGA stop codon in cells lacking SNORA24, compared to isogenic control cells (presented in new Figure 5B). In contrast, no difference in readthrough of an UAG stop codon was detected in the same cells.

Altogether, these findings support a role for SNORA24-guided modifications in translation accuracy, that appears to dependent, in part, on the specific codon been decoded.